# Revisiting the Moisture Budget of the Mediterranean Region in the ERA5 Reanalysis

Roshanak Tootoonchi[1], Simona Bordoni[1], and Roberta D'Agostino[2]

[1]Department of Civil, Environmental and Mechanical Engineering, University of Trento, Italy
[2]National Research Council, Institute of Atmospheric Sciences and Climate, Lecce, Italy

**Correspondence:** Roshanak Tootoonchi (roshanak.tootoonchi@unitn.it)

**Abstract.**

Moisture budget assessments from reanalyses and climate models have provided fundamental insights into the maintenance and response to perturbations of the hydrological cycle in the Mediterranean region. Here we perform similar analyses using the latest generation European Centre for Medium-Range Weather Forecasts (ECMWF) Re-Analysis ERA5, and we complement
previous work by further decomposing the mean flow into contributions by the zonal-mean flow, which is dominated by the mean meridional circulation, and by zonally anomalous circulations and/or moisture, namely the stationary eddies. According to ERA5, in the annual mean, net evaporation (negative $P - E$) over the ocean and net precipitation (positive $P - E$) over land are primarily due to submonthly transient eddies converging moisture originating from the ocean into the surrounding land. Overall, total stationary eddies reinforce the transient tendency over the ocean but oppose it over land, with the zonal-mean
flow exerting a minor drying tendency limited to the region's southernmost latitudes. The total stationary eddy moisture flux divergence arises from a strongly divergent zonally anomalous circulation acting on the zonal-mean moisture. This is partly opposed by the pure stationary eddy term, which provides moisture flux convergence through both divergence and advection of zonally anomalous moisture by the zonally anomalous circulation. The relative magnitude of these terms changes over the seasonal cycle, explaining the transition from net precipitation during winter (DJF) to net evaporation during summer (JJA)
over land. More specifically, as transient eddies weaken during the warm season, the strengthened divergent total stationary eddy moisture flux becomes dominant and causes strong drying and negative net precipitation. Somewhat surprisingly, moisture flux divergence by the mean meridional circulation is found to play a minor role in the Mediterranean region across all seasons except autumn (SON).

## 1 Introduction

The Mediterranean climate is broadly defined as a temperate climate occurring from subtropical to lower middle latitudes characterized by dry, hot summers and moderately wet winters. Regions experiencing this climate type include California and southwestern Australia, as well as the Mediterranean region itself, from which the name for this type of climate originates. In the Mediterranean, midlatitude winter storms provide the moisture flux convergence sustaining the cold-season precipitation.

Thanks to the moderate subtropical temperatures, precipitation is primarily in the form of rainfall, except at high elevations over the Alps and other mountain ranges (Şahin et al., 2015).

In summer, the region is under the influence of strong subsidence, which gives rise to stable conditions, clear skies, and precipitation minima (D'Agostino and Lionello, 2020). This strong seasonal cycle leads to an imbalance between precipitation and evaporation, with the latter exceeding the former, and a climate that can be categorized as semi-arid (Seager et al., 2014a, b).

The strong seasonality in the hydrological cycle and the resulting water scarcity in all Mediterranean-type climate regions explain the high concern for their water resources (Giorgi and Lionello, 2008; Zappa et al., 2015; Seager et al., 2014a). For instance, the climate of the Mediterranean region is distinctively sensitive to the spatial and temporal characteristics of winter storms as they move into the region (Şahin et al., 2015; Lionello and Scarascia, 2018). The quantity and intensity of the unique and irregular pattern of associated winter rainfall (D'Agostino and Lionello, 2020) is significantly influenced by the intensity and position of the North Atlantic storm track and the rapid development of extratropical cyclones across the region (Lionello and Scarascia, 2018; Ukhurebor et al., 2022). Even relatively minor perturbations in the general circulation of the atmosphere, such as shifts in the location of midlatitude storm tracks or subtropical highs, both due to natural variability and anthropogenic climate change, can lead to substantial changes in the Mediterranean climate (Mariotti et al., 2002; Giorgi and Lionello, 2008; D'Agostino et al., 2020). Not surprisingly, many studies have identified the Mediterranean region as a climate-change "hotspot" marked by temperatures increasing faster than the globally averaged temperature and a propensity for increased aridity (Seager et al., 2014a; Byrne and O'Gorman, 2015; Lionello and Scarascia, 2018; D'Agostino and Lionello, 2020).

Previous work has leveraged global reanalyses and several diagnostics to understand how the hydrological cycle is maintained both at the global and regional scales. Among those, the atmospheric moisture budget has been particularly powerful, as it has allowed for an understanding of how convergence of moisture flux by atmospheric circulations leads to net precipitation (that is, the difference between precipitation and evaporation, $P - E$) (e.g., Seager et al., 2007; Li et al., 2013; Wills and Schneider, 2015; Şahin et al., 2015; Wills et al., 2016; D'Agostino et al., 2020; Minallah and Steiner, 2021). The same budget has also been applied to the decomposition of future precipitation projections within the Coupled Model Intercomparison Project (CMIP) model archives into a more robust and theoretically constrained thermodynamic component and a more uncertain and less understood dynamic component, both at the global and regional scales (e.g., Held and Soden, 2006; Byrne and O'Gorman, 2015; Seager et al., 2010, 2014a, b; Elbaum et al., 2022).

In a seminal contribution to the understanding of the maintenance of the hydrological cycle in the Mediterranean region, Seager et al. (2014a) studied its atmospheric moisture budget within the European Centre for Medium-Range Weather Forecasts (ECMWF) ERA-Interim reanalysis and CMIP5 climate model simulations. By decomposing the moisture flux convergence into contributions from the monthly mean flow and from eddying motions, i.e., variations from the temporal monthly mean (see Fig. 4d, 4e, 4f), they showed how convergence by submonthly transient eddies, partially offset by divergence by the time-mean flow, sustains positive net precipitation over land areas in winter (November to April). Şahin et al. (2015) further showed that the transient eddy moisture flux originating from the North Atlantic Ocean sustains wet conditions in western and central Europe in winter. Contrarily, the contribution from transient storms over the Mediterranean basin weakens in summer due to weaker thermal contrasting properties and resulting less-frequent frontal activities.

One thing that appears to be missing in the literature for a more detailed understanding of the Mediterranean hydrological cycle is, however, separating the time mean into contributions from the time- and zonal-mean flow (the mean meridional circulation) and from stationary eddies. It is clear that the zonally symmetric mean-flow moisture flux in lower latitudes is associated with subsidence within the descending brand of the Hadley cell (Held and Soden, 2006; Byrne and O'Gorman, 2015; Wills and Schneider, 2015; D'Agostino and Lionello, 2020). However, the extent to which the subsidence over the Mediterranean region is linked to the zonal-mean circulation in both annual and seasonal means remains somewhat unclear. Variations about the zonal mean due to surface forcing or stationary eddies are expected to make a significant contribution to the spatial variability in the moisture flux and its convergence (Chen and Bordoni, 2014; Wills and Schneider, 2015; Wills et al., 2016), as it is well known that orography and land-sea heating contrast are major factors that force extratropical stationary waves (Held et al., 2002; Wills and Schneider, 2015). Orographically forced stationary waves can lead to low-level divergence and the formation of subtropical midlatitude dry zones, as well as low-level stationary eddy mass convergence associated with wet zones (Broccoli and Manabe, 1992). Specific to the Mediterranean region, enhanced summer subsidence has been related to a forced response to Asian monsoon heating, amplified by topography in the Middle East (Rodwell and Hoskins, 1996, 2001; Simpson et al., 2015).

Several studies have exposed and quantified the impact of low-frequency stationary waves on both regional and global hydroclimate variability. For instance, Chen and Bordoni (2014) used observations and general circulation model (GCM) simulations to explore the influence of the Tibetan Plateau on the East Asian Summer Monsoon, a quasi-stationary subtropical rainfall band, by analysing and decomposing the moisture and moist static energy budgets into zonal-mean, stationary, and transient eddy contributions. Wills and Schneider (2015) used reanalysis data to understand spatial variations in global net precipitation via decomposition of the atmospheric moisture budget into the zonal mean, which accounts for 40% of the spatial variability in global $P - E$, and variations about the zonal-mean hydrological cycle, which explains the remaining 60%. Importantly, this study clarified the leading-order role that zonally anomalous circulations play in the observed zonal variations of net precipitation, with zonally anomalous specific humidity providing a more secondary contribution.

In this paper, we aim to fill the existing knowledge gap. More specifically, we examine the maintenance of the hydrological cycle in the Mediterranean in the 5th and latest generation ECMWF reanalysis (ERA5), and we complement previous work by further decomposing the mean flow into contributions by the zonal-mean flow and by zonally anomalous circulations and/or moisture, namely the stationary eddies. As shown below, this allows us to clarify existing confusion on the impact, or lack thereof, of zonal-mean descent within the subtropical branch of the Hadley cell on the region's aridity and more directly assess the relative role of zonal asymmetries in circulation and moisture. Specific goals of this work are:

1. To expose any climatological trends that might be present in the ERA5 dataset;

2. To assess the contribution of mean flow, transient, and stationary eddies to the maintenance of net precipitation in the annual and seasonal means over land and sea within the broad Mediterranean region;

3. To explore how moisture flux convergence by stationary eddies arises from zonal variations in wind patterns and moisture content.

The paper is organized as follows: We first describe data and methods in Section 2.1, with details on the moisture budget decomposition in Section 2.2 and moisture budget global and regional closure in ERA5 in Section 2.3. Results of our analyses for the annual and seasonal means are discussed in Sections 3.1 and 3.2, respectively. We summarize our results in Section 4 with some discussion of their relevance to future studies on climate change in the Mediterranean.

## 2  Data and Methods

### 2.1  ERA5

The atmospheric moisture budget of the Mediterranean region is studied using the atmospheric ERA5 reanalysis dataset, which currently covers the time period from 1940 to present. The ERA5 atmospheric archive provides global data that have been regridded to a reduced Gaussian grid with a horizontal resolution of $0.25° \times 0.25°$ ($\sim 27 - 28$ km), at 1-hourly temporal resolution, and on 137 vertical levels up to the 0.01 hPa pressure level (Hersbach et al., 2020; Mayer et al., 2021). In this study, we use monthly averaged data on pressure levels as well as column-integrated mass-consistent atmospheric energy and moisture budget monthly data derived from 1-hourly ERA5 reanalysis data (Copernicus Climate Change Service, 2022) from 1979 to present archived by the Copernicus Climate Data Store (CDS). ERA5 publishes vertically integrated eastward and northward moisture fluxes, as well as the divergence of these fluxes, that employ mass-consistent horizontal wind fields (Copernicus Climate Change Service, 2022), which is what we use in this study for the total moisture flux convergence. It is worth mentioning that ERA5 also publishes moisture fluxes and associated divergence at hourly resolution, but these do not employ mass-consistent horizontal wind fields and are contaminated with numerical noise. While the high-frequency output would allow us to compute explicitly the transient eddy contribution, we decided against using these fluxes because they do not satisfy global properties such as having a zero global mean divergence and are affected by numerical noise (Gutenstein et al., 2021; Mayer et al., 2021).

The moisture budget calculations were done over the $1979 - 2020$ time period; different subsets within this period were checked to test the robustness of the results. The primary objective of this study is to understand the role of moisture transport by stationary eddies and synoptic-scale eddies over a broad region of considerable orographic complexity, encompassing the Mediterranean Sea and adjacent land (from 10°W to 40°E and from 30°N to 50°N). Notice how this includes regions not traditionally identified as belonging to the Mediterranean, but the focus on this broader box allows for the analysis of larger-scale patterns and possibly distinct behaviors of the Mediterranean with respect to other neighbouring areas. To eliminate noise on small spatial scales, monthly ERA5 data with the original resolution of $0.25° \times 0.25°$ were regridded to $1° \times 1°$ spatial resolution.

### 2.2  Atmospheric Moisture Budget Decomposition

We analyse the moisture budget of the Mediterranean region to evaluate the contributions from the mean flow, transient, and stationary eddies in the maintenance of the observed climatological net precipitation patterns in the annual and seasonal means

over land and ocean. ERA5 provides data on pressure levels, and thus we study the budget in pressure coordinates. Given that

$P$ and $E$ are the only fundamental sink and source of water vapor in an atmospheric column (e.g., Wills and Schneider, 2015), under steady state, their difference must be equal to the convergence of water vapor flux (Trenberth et al., 2007):

$$\overline{P} - \overline{E} = -\nabla \cdot \langle \overline{\mathbf{u}q} \rangle \tag{1}$$

where

$$\langle \cdot \rangle = \int\limits_{0}^{\overline{Ps}} \cdot \frac{dp}{g} \tag{2}$$

$\langle \cdot \rangle$ represents a mass-weighted vertical integral, $\overline{(\cdot)}$ a monthly time mean, $\nabla$ is the nabla operator on the sphere, $\mathbf{u}$ is the horizontal wind vector, $q$ is the specific humidity, $Ps$ is the surface pressure, and $g$ is the gravitational acceleration. All three terms are directly available from ERA5, and, as mentioned above, we use the readily available divergence of the vertical integral of water vapor flux that employs mass-consistent horizontal wind to calculate the total moisture flux convergence.

Following previous studies (Seager et al., 2010; Wills and Schneider, 2015; D'Agostino and Lionello, 2020), we use

Reynolds' decomposition to decompose the total moisture flux into contributions from the time mean (with monthly averages shown by overbars) and contributions from the transient eddies (shown by prime), such that $(\cdot) = \overline{(\cdot)} + (\cdot)'$. This decomposition of the budget requires vertical integration of different flux components, for which we select eleven pressure levels (1000 950 900 850 750 650 500 300 200 100 50 hPa) for computational efficiency. Results shown below are robust to a different choice, including a higher number, of selected pressure levels. Eq. 1 hence becomes:

$$\overline{P} - \overline{E} = -\nabla \cdot \langle \overline{\mathbf{u}} \, \overline{q} + \overline{\mathbf{u}'q'} \rangle \tag{3}$$

The first term on the right is the moisture flux convergence (MFC) by the mean-flow contribution, and the second term is the MFC by the transient eddies. Climatological annual and seasonal averages of all terms are then computed and shown in Section 3.

We are interested in further decomposing the budget into zonal mean (shown by brackets) and variations from the zonal

mean that represent the stationary components (shown by stars), such that $(\cdot) = [\cdot] + (\cdot)^*$. With this further decomposition, the budget becomes:

$$\overline{P} - \overline{E} = -\nabla \cdot \langle [\overline{\mathbf{u}}][\overline{q}] \rangle - \nabla \cdot \langle \overline{\mathbf{u}}^* \overline{q}^* + [\overline{\mathbf{u}}]\overline{q}^* + \overline{\mathbf{u}}^*[\overline{q}] \rangle - \nabla \cdot \langle \overline{\mathbf{u}'q'} \rangle \tag{4}$$

where

$$\overline{\mathbf{u}}^* \overline{q}^* + [\overline{\mathbf{u}}]\overline{q}^* + \overline{\mathbf{u}}^*[\overline{q}] \equiv \overline{\mathbf{u}}^\dagger \overline{q}^\dagger \tag{5}$$

In Eq. 4, the first term on the right is the MFC by the zonal-mean circulation, and the second term on the right is the total MFC by stationary eddies $\overline{\mathbf{u}}^\dagger \overline{q}^\dagger$ (Eq. 5). Only the first of the three terms on the left-hand side of Eq. 5 represents the contribution

from the classical (pure) stationary eddy flux, $\overline{\mathbf{u}}^*\overline{q}^*$, to the total MFC, which includes correlations of zonally anomalous circulations $\overline{\mathbf{u}}^*$ and zonally anomalous moisture $\overline{q}^*$. The other two terms are the cross terms, which capture interactions of stationary eddies with zonal means. Note how only the pure stationary eddy term survives if one considers the global zonal mean (i.e., $[\overline{\mathbf{u}}^*\overline{q}^*] \equiv [\overline{\mathbf{u}}^\dagger\overline{q}^\dagger]$). While not contributing to the global zonal mean, the cross terms have, however, been shown to be very important at the regional scale (Kaspi and Schneider, 2013; Wills and Schneider, 2015).

Several studies have pointed out the inevitable introduction of errors in the analyses of the vertically integrated moisture budget, which results in possibly non-zero residuals (e.g., Seager and Henderson, 2013; Wills and Schneider, 2015; Minallah and Steiner, 2021). Differentiation approximations, spatial and temporal discretization, and vertical interpolation all contribute to information loss and errors. For these reasons, we compute all other terms explicitly, but estimate the transient eddy term implicitly as the moisture budget residual, that is:

$$\nabla \cdot \langle \overline{\mathbf{u}'q'} \rangle = -(\overline{P} - \overline{E}) - \nabla \cdot \langle [\overline{\mathbf{u}}][\overline{q}] \rangle - \nabla \cdot \langle \overline{\mathbf{u}}^*\overline{q}^* + [\overline{\mathbf{u}}]\overline{q}^* + \overline{\mathbf{u}}^*[\overline{q}] \rangle \tag{6}$$

Results from the decomposition as per Eq. 4 are shown in Figs. 4, Figs. 10 and 11 for the annual and seasonal means, respectively. It is important to note that up to this point, following Byrne and O'Gorman (2015), we take the divergence of the vertically integrated fields in the zonal and meridional direction rather than vertically integrating the moisture flux divergence fields, as alternatively proposed by other studies (e.g., Seager et al., 2010, 2014a, b; Wills and Schneider, 2015; D'Agostino and Lionello, 2020). Hence, there is no term related to the surface-pressure gradient (e.g., Seager et al., 2014a). To get further physical insight into the MFC due to stationary eddies, we however further decompose the associated terms in contributions from wind divergence and advection, that is

$$\nabla \cdot \langle \overline{\mathbf{u}}^*\overline{q}^* + [\overline{\mathbf{u}}]\overline{q}^* + \overline{\mathbf{u}}^*[\overline{q}] \rangle = \langle (\overline{q}^*\nabla \cdot \overline{\mathbf{u}}^* + \overline{\mathbf{u}}^* \cdot \nabla\overline{q}^*) + (\overline{q}^*\nabla \cdot [\overline{\mathbf{u}}] + [\overline{\mathbf{u}}] \cdot \nabla\overline{q}^*) + ([\overline{q}]\nabla \cdot \overline{\mathbf{u}}^* + \overline{\mathbf{u}}^* \cdot \nabla[\overline{q}]) \rangle + \overline{S} \tag{7}$$

In so doing, the nabla operator is taken inside the integral, and the surface-pressure gradient term, $\overline{S}$, does appear (Seager and Henderson, 2013; Seager et al., 2014a; Wills and Schneider, 2015). We evaluate this term as the residual between the left-hand side and the sum of the advection and divergence contributions on the right-hand side of Eq. 7 and find it to be negligible. Results from this further breakdown of the MFC by the stationary eddy terms are shown in Figs. 7 and 8.

We are also interested in quantifying which components in Eq. 4 have a bigger contribution to the zonal-mean pattern of $P-E$ within the Mediterranean. Hence, following Wills and Schneider (2015), we write the zonal sector-mean moisture budget as:

$$\{\overline{P}\} - \{\overline{E}\} = -\{\nabla \cdot \langle [\overline{\mathbf{u}}][\overline{q}] \rangle\} - \{\nabla \cdot \langle \overline{\mathbf{u}}^\dagger\overline{q}^\dagger \rangle\} - \{\nabla \cdot \langle \overline{\mathbf{u}'q'} \rangle\} \tag{8}$$

where $\{\cdot\}$ denotes an average in longitude over the broad Mediterranean region (that is, from $10°$W to $40°$E). This wide sector was chosen to be consistent with previous work (e.g., Giorgi and Lionello, 2008; D'Agostino and Lionello, 2020; Tuel et al., 2021) and to include the Mediterranean Sea as well as all surrounding land regions. Patterns discussed below (Figs. 3 and 9) are robust to small changes in the selected longitude range.

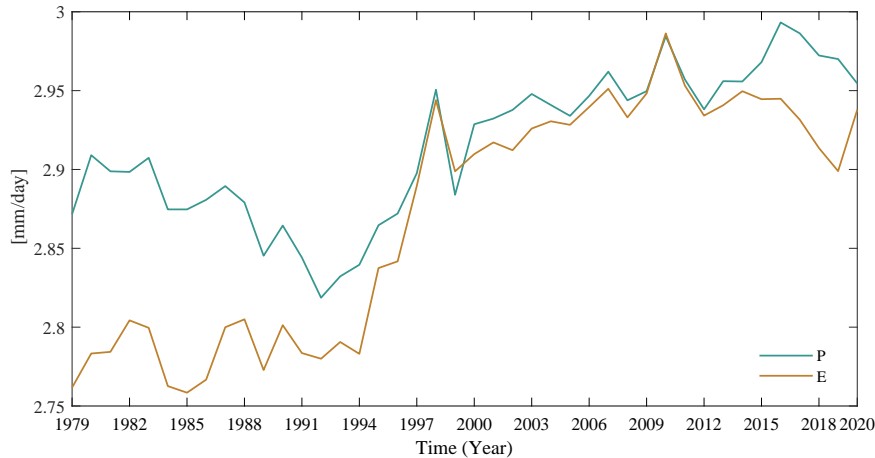

**Figure 1.** Temporal evolution of global annual-mean $P$ and $E$ from ERA5 in the $1979 - 2020$ period.

## 2.3 Atmospheric Moisture Budget Closure

Before delving into the maintenance of the Mediterranean moisture budget, we analyse the closure of the global moisture budget in ERA5, that is, we explore if and to what extent globally averaged $P$ is equal to globally averaged $E$ in the $1979 - 2020$ period under consideration. Fig. 1, which shows the evolution of these two quantities averaged globally and annually, reveals a few concerning results. Firstly, globally averaged $P$ always exceeds globally averaged $E$ by an average value of 0.045 mm/day over the $1979 - 2020$ period, suggesting there exists an artificial moisture source in ERA5. Secondly, the imbalance between the two fields does not remain constant over time but decreases significantly after 1995. In particular, while the excess of globally averaged $P$ over globally averaged $E$ is about 0.083 mm/day in the $1979 - 1995$ period, it decreases to 0.015 mm/day after 1995. Previously, Mayer et al. (2021) reported analogous results from ERA5, with residual values of about 0.03 mm/day in the $1985 - 2018$ period and of about 0.02 mm/day in the $2000 - 2018$ period. These changes in the residual do not arise from departures from steady state, an assumption that remains extremely well verified throughout our analysis period. In fact, similar discontinuities in precipitation from ERA5 and other global reanalyses have been reported in Scoccimarro et al. (2024) and attributed to the progressive addition of satellite data sources (Diniz and Todling, 2020).

While for our purposes a closed moisture budget would be desirable, but not strictly necessary, improvements over time introduce trends in both $E$ and $P$ that appear unrealistic and might not reflect observed changes. Specifically, from the mid-1990s, $E$ increases more drastically than $P$, and this seems to be the main change that results in a better balance (Hersbach et al., 2020). This highlights how extreme caution should be taken when analysing any trend in quantities related to the ERA5 moisture budget.

If we now consider the moisture budget averaged over the Mediterranean region, $P - E$ is negative on the annual average. This highlights the known characteristics of this basin: the Mediterranean Sea acts as a source of moisture, supporting both

local recycling and precipitation over neighboring land regions. A negative total MFC (i.e., divergence of moisture flux, see Eq. 1) must balance the net evaporation (negative $P - E$), which is what we see in Fig. 2. However, as was the case for the global mean, the regional moisture balance is not closed, especially in the first 10 years of the time period under consideration. Total MFC, with a climatological annual mean value of $-0.503$ mm/day, exceeds $P - E$, with a climatological annual mean value of $-0.367$ mm/day. There is a decreasing trend in $P - E$ from 1979 to 2020, which is due to an increasing trend in $E$ and a rather steady $P$ (not shown). Yet, there seems to be no statistically significant trend in total MFC. All in all, the budget over the Mediterranean is not strictly closed, and the imbalance is larger over land (Fig. 2b). Mirroring what was seen at the global scale, the closure of the regional moisture budget, which does not arise from steady-state departures, improves over time, with a decreasing imbalance between $P - E$ and total MFC (Fig. 2a).

What can we conclude about hydroclimate changes in the Mediterranean region in the past 40 years, according to ERA5? Can we trust the signal? While it is possible to choose different time windows and find some trends, the "artificial jumps" in global $P$ and $E$ from ERA5 make it difficult to extract any meaningful trends or changes in the hydroclimate, because of remaining inconsistencies in large-scale budgets of moisture and energy (Hersbach et al., 2020; Allan et al., 2020; Mayer et al., 2021). We hence caution against the use of ERA5 for trend identification and attribution. However, like other reanalysis products, it remains a very useful tool to study mechanisms responsible for the maintenance of climatic patterns both at global and regional scales, as we do in this study.

## 3 Results

### 3.1 The Annual Mean Mediterranean Moisture Budget

We begin by analysing how the long-term, annually averaged moisture budget over the Mediterranean region is maintained in ERA5 in the zonal sector mean, and we then explore the spatial patterns of all relevant terms. The zonal sector-mean decomposition of the moisture budget, following Eq. 8, in our analysis region is shown in Fig. 3. Similarly to what is seen in the global zonal mean (e.g., Wills and Schneider, 2015), the zonally averaged net precipitation in the Mediterranean region transitions from net evaporation to net precipitation at around 43°N, slightly poleward of the globally zonally averaged value of 37°N (not shown). Transient eddies are primarily responsible for the associated poleward moisture transport by diverging moisture from the lower midlatitudes and converging it to the higher latitudes. Note how the net precipitation minimum at 35°N coincides with the transient eddy moisture flux divergence maximum around the same latitude band. While smaller in magnitude, total stationary eddies and the mean meridional circulation provide net moisture flux divergence at all latitudes. South of 32°N, their combined effect is large enough to more than offset the moisture flux convergence by transient eddies and to maintain net evaporation on the southern edge of the Mediterranean region.

While illustrative of the role that different atmospheric circulations play in meridional moisture transport and the resulting zonally averaged net precipitation patterns, the zonal-mean analysis hides large zonal variability and land-sea contrast and possibly obscures the role of zonally varying circulations in their maintenance. The spatially varying moisture budget and its components are hence shown in Figs. 4.

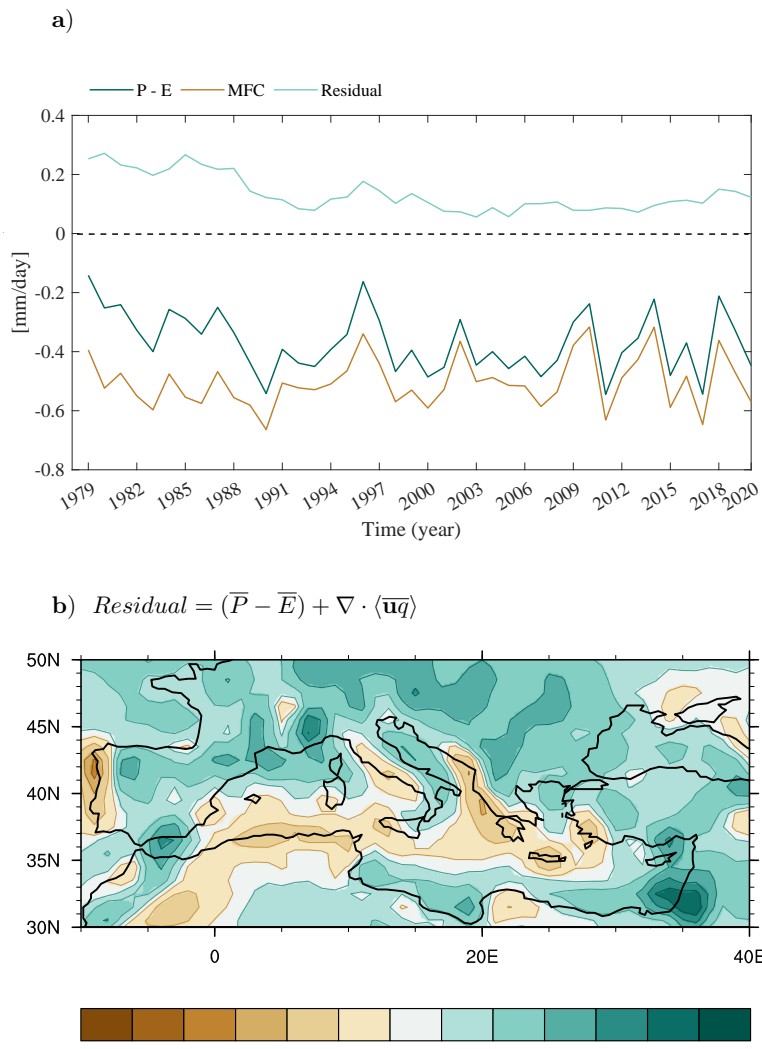

**Figure 2.** Annual mean moisture budget components in the Mediterranean region from ERA5 in the $1979 - 2020$ period: (a) temporal evolution of $P - E$ (green line), total moisture flux convergence ($MFC$, brown line), and the difference between the two (residual, cyan line), (b) time-averaged spatial variations of the imbalance between $P - E$ and total $MFC$ in millimeters per day. The total MFC is obtained from the publicly published mass-consistent divergence of the vertical integral of moisture flux. See Section 2.1 for more details.

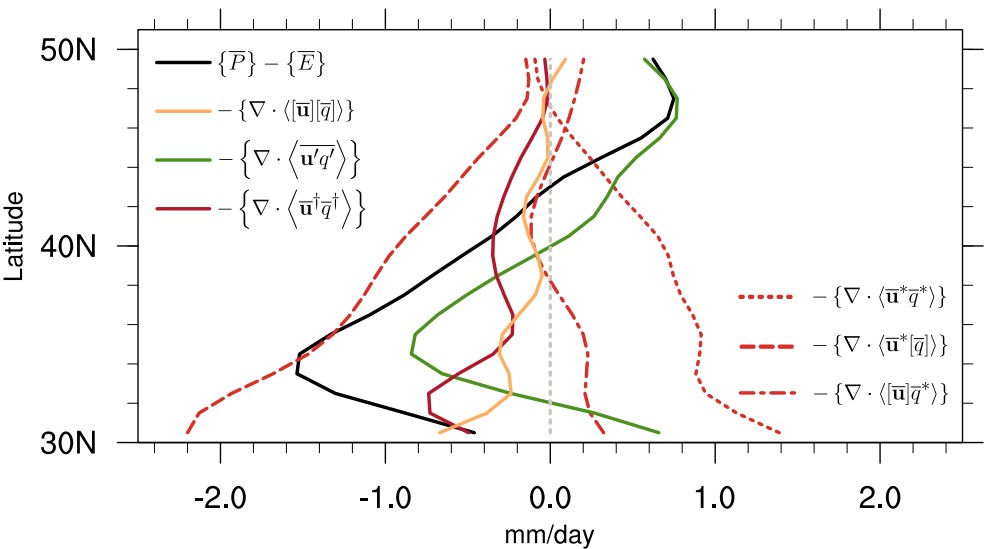

**Figure 3.** Sector-mean annually averaged (ANN) climatological Mediterranean moisture budget components in the $1979 - 2020$ period from ERA5. Each curve is smoothed in latitude using a 3-point running average filter.

All northern Mediterranean land areas, such as the Iberian Peninsula, France, the Alps, the Balkans, and Turkey, have an excess of $P$ over $E$ (Fig. 4a). Annually averaged positive net precipitation is particularly strong in high-topography regions (the Alps and Balkans). In agreement with previous work (Seager et al., 2014a), over the Mediterranean Sea, $E$ exceeds $P$, confirming how this ocean basin acts as a source of moisture for the surrounding land masses. Evaporation appears particularly

dominant in the eastern Mediterranean Sea, due to both strong near-surface 900-hPa winds and warm sea surface temperatures (not shown).

According to ERA5, positive $P - E$ over all land regions surrounding the Mediterranean Sea, with the exception of northern Africa, is predominantly maintained by the sub-monthly transient eddy moisture flux that converges over these regions moisture originating from the sea (Fig. 4d). The mean flow partially counteracts the wettening tendency of the transient eddies over land

regions, with the exception of northern Africa, and reinforces it over the ocean, except for the Adriatic Sea (Fig. 4b). Over northwestern Africa, where the time-mean flow is convergent, the transient eddies are strongly divergent, which points to the opposing behaviour of the two phenomena leading to negligible net precipitation in the region. This counterbalancing contribution by the mean flow and transient eddies is consistent with earlier work based on the previous generation of the ECMWF reanalysis, ERA-Interim, (Seager et al., 2014a), giving us confidence in the robustness of our results.

The time-mean moisture flux convergence can be further decomposed into contributions by the zonally averaged circulation and the stationary eddies (Eq. 4). The long-term averaged zonal-mean circulation, dominated by the mean meridional circula-

tion, results in divergence of moisture from the subtropical southern edge of the Mediterranean region (from 30°N to 36°N, Fig. 4c). Everywhere else in the region, the contribution by the zonal-mean moisture flux is negligible, revealing how moisture flux and the associated convergence by stationary eddies dominate the time-mean flow, pointing to the immense impact of variations about the zonal mean on the hydrological cycle in the Mediterranean region.

As mentioned before, the total stationary eddy moisture flux arises from three different components that contain stationary terms (see Eq. 5). We are particularly interested in knowing which component, if any, contributes the most to the hydrological cycle of the region.

According to ERA5, pure stationary eddies, $-\nabla \cdot \langle \overline{\mathbf{u}}^* \overline{q}^* \rangle$, converge moisture over the Mediterranean land region as well as the sea, except for the southwestern Mediterranean (Algeria) and western Black Sea (see Fig. 4f). The transport of zonally symmetric moisture by the zonally anomalous circulation, $-\nabla \cdot \langle \overline{\mathbf{u}}^* [\overline{q}] \rangle$, results in moisture divergence from the Mediterranean region, including the Mediterranean Sea, with the exception of northwest Africa, northern Italy, and the Adriatic Sea (Fig. 4g). The divergence is the strongest over the eastern Mediterranean, where the zonal variations in circulation are the largest. As discussed more in detail below, these patterns primarily mirror patterns of the zonally anomalous lower-level wind divergence. The transport of zonally anomalous moisture by the zonally symmetric circulation, $-\nabla \cdot \langle [\overline{\mathbf{u}}] \overline{q}^* \rangle$, yields a different result: convergence of moisture over the Mediterranean Sea, northern Africa, and western Black Sea, and divergence over the western Mediterranean (i.e., the Iberian Peninsula, Morocco, etc.), Corsica, the Tyrrhenian Sea, and Turkey (Fig. 4h).

These results raise the question as to why these different stationary terms contribute the way they do to net precipitation in the Mediterranean region. In addition to understanding the spatial patterns of all stationary MFC terms, we are also interested in explaining the partial cancellation between contributions arising from the transport of zonal-mean moisture by the zonally anomalous circulation ($-\nabla \cdot \langle \overline{\mathbf{u}}^* [\overline{q}] \rangle$, Fig. 4g) and the pure stationary eddies ($-\nabla \cdot \langle \overline{\mathbf{u}}^* \overline{q}^* \rangle$, Fig. 4f).

A good starting point is to examine patterns of zonally asymmetric moisture and circulation individually, shown in Figs. 5 and 6. Note how in these two figures we use a more extended domain than the one used elsewhere to link these analyses to previously published results, as discussed below. Considering that moisture is primarily concentrated at low levels, Fig. 5 shows variations from the zonal mean and the zonal mean of specific humidity at the 850-hPa representative level ($\overline{q}_{850}^*$, Fig. 5a, and $[\overline{q}_{850}]$, Fig. 5b, respectively). Low-level moisture over the southern Mediterranean region and the Mediterranean Sea itself is smaller than the global zonal mean (Fig. 5a), resulting in negative (dry) zonal anomalies over these areas. No clear land-sea contrast is seen in these anomalous humidity patterns; rather, the signal is one of a meridional contrast between a northern moister ($\overline{q}_{850}^* > 0$) part and a southern drier ($\overline{q}_{850}^* < 0$) part. An exception to this overall behavior is northwestern Africa, where $\overline{q}_{850}^*$ is positive. Not particularly informative, Fig. 5b shows the expected pattern of decreasing zonal-mean specific humidity ($[\overline{q}_{850}]$) with latitude.

Characterizing zonally asymmetric circulation patterns that influence net precipitation is less straightforward, as stationary eddies can impact moisture flux convergence through both horizontal advection and vertical motion (horizontal divergence). Given the well-established importance of vertical motions on large-scale zonally anomalous $P - E$ patterns (e.g., Wills and Schneider, 2015), Fig. 6 shows variations from the zonal mean and the zonal mean of the 500-hPa vertical pressure velocity ($-\overline{\omega}_{500}^*$, Fig. 6a, and $-[\overline{\omega}_{500}]$, Fig. 6b, respectively), which well correlates with low-level divergence and is not affected by

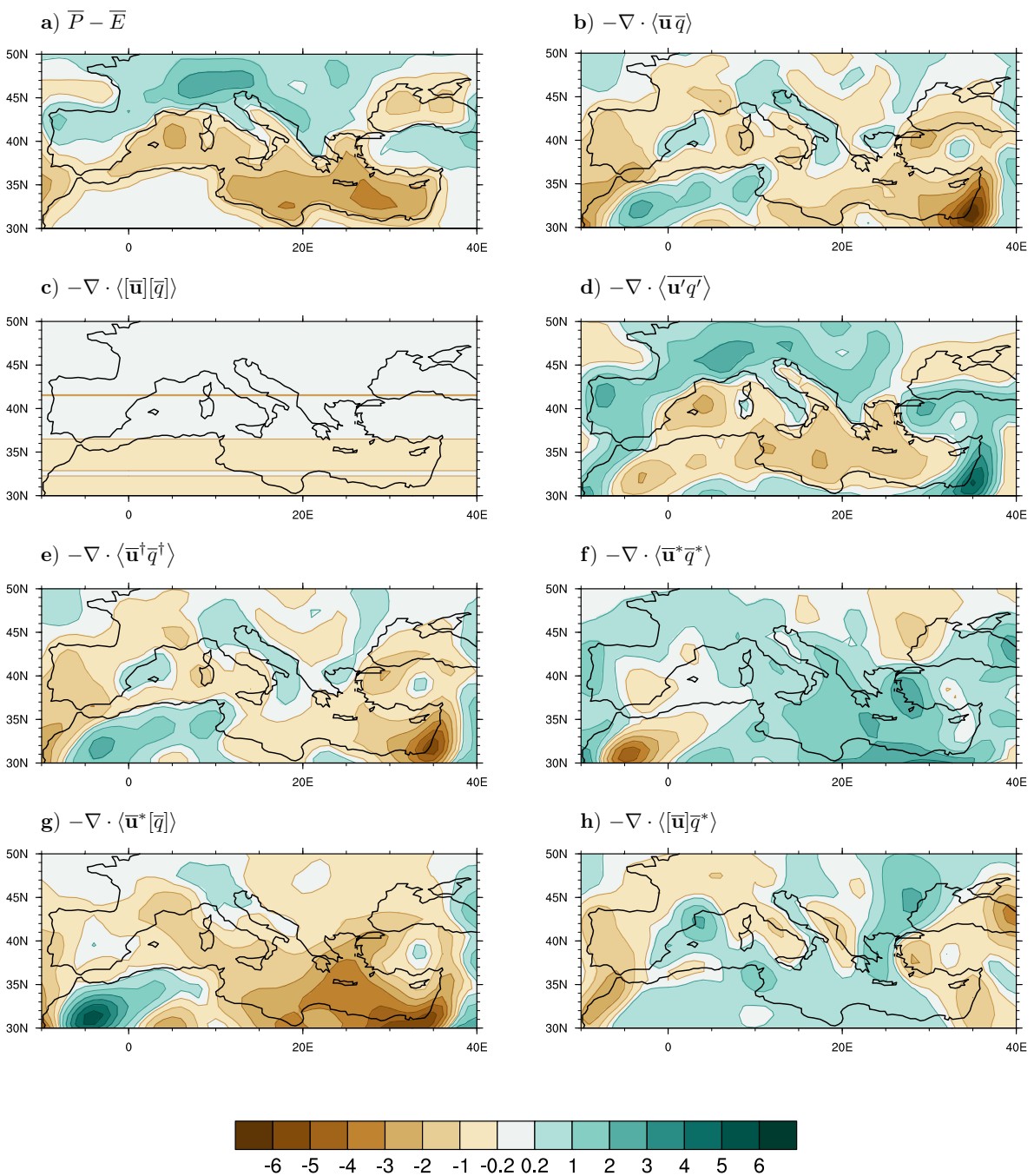

**Figure 4.** Annual mean climatological Mediterranean moisture budget in the $1979-2020$ period from ERA5: (a) $P-E$, MFC due to (b) monthly mean flow, (c) zonal mean circulation, (d) transient eddies, (e) total stationary eddies, and its three components arising from (f) pure stationary eddies, (g) transport of zonal mean moisture by the zonally anomalous circulation, and (h) transport of zonally anomalous moisture by the zonal mean circulation. All fields are smoothed using a 3×3 point smoothing filter. Units are millimeters per day.

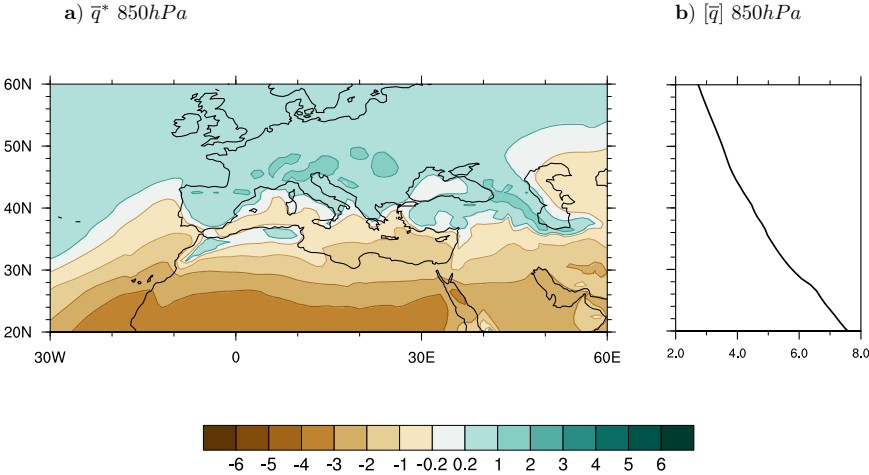

**a)** $\bar{q}^*$ $850hPa$

**b)** $[\bar{q}]$ $850hPa$

**Figure 5.** Annual mean climatological low-level (850 hPa) (a) zonally anomalous moisture and (b) zonal-mean moisture from ERA5 in the $1979 - 2020$ period. Units are $\mathrm{g\,kg^{-1}}$.

numerical noise. Please note how in Fig. 6 we show the negative vertical pressure velocity, so that positive (negative) values indicate ascending (descending) motion. At 500 hPa, the zonally anomalous vertical velocity ($-\overline{\omega}^*_{500}$, Fig. 6a) features strong descent across the Mediterranean region, except northwestern Africa, where the Atlas mountain range is located, and the northern Balkans. Anomalous ascending motion can be seen in the Middle East over the Zagros mountains, with enhanced subsidence to both their east and west, the latter extending over the central and eastern Mediterranean. Maximum subsidence is centered over the eastern Mediterranean. This is in strong agreement with previous published papers, which link the subsidence over the Mediterranean region to the influence of topography, more specifically the Zagros mountains (Simpson et al., 2015) and the Atlas mountains (Rodwell and Hoskins, 1996), and its modulation of subsidence induced by the Indian monsoon heating (Rodwell and Hoskins, 1996; Cherchi et al., 2014, 2016; Simpson et al., 2015). The global zonal-mean vertical velocity ($-[\overline{\omega}]$, Fig. 6b) instead reveals a region of negligible ascending motion between 40°N and 50°N, flanked by descent to its south and weak ascent to its north.

To the extent that horizontal advection is negligible, MFC by the three stationary eddy terms, at least in terms of overall spatial patterns, can be understood by considering how the zonally averaged and/or the zonally anomalous vertical motion act on the zonally averaged and/or the zonally anomalous moisture to determine patterns of vertical moisture advection (see also Wills and Schneider, 2015). A comparison of Fig. 4g and Fig. 6a reveals how patterns of convergence of zonal-mean moisture by the zonally anomalous circulation, $-\nabla \cdot \langle \overline{\mathbf{u}}^* [\overline{q}] \rangle$, are well explained by patterns of the zonally anomalous mid-tropospheric vertical velocity, $-\overline{\omega}^*_{500}$, with moisture divergence (convergence) coinciding with regions of subsiding (ascending) motion. In

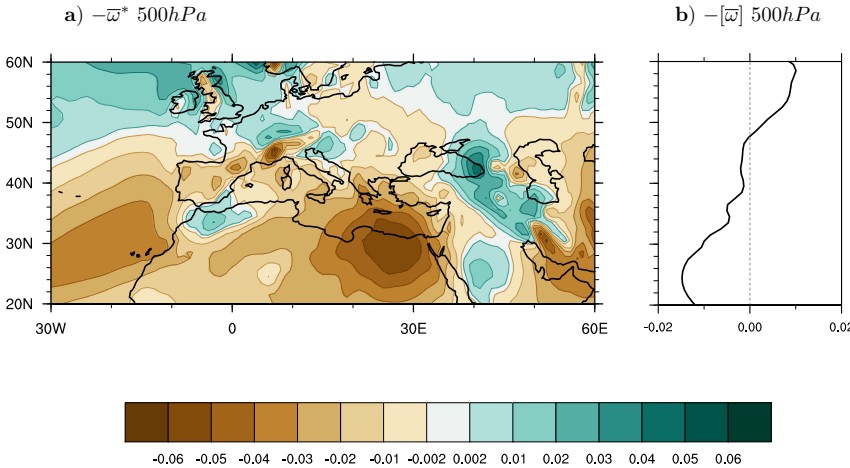

**Figure 6.** Annual mean climatological mid-tropospheric (500 hPa) (a) anomalous vertical velocity and (b) zonal-mean vertical velocity from ERA5 in the $1979-2020$ period. Units are $\mathrm{Pa\,s}^{-1}$. Note how here we show negative pressure velocity; hence, positive values (green) denote ascending motion and negative values (brown) denote descending motion.

other words, the leading-order effect of this term is due to zonal variations in vertical motion and associated horizontal wind convergence, which is primarily descending and diverging moisture away from the Mediterranean region.

The zonally anomalous descending motion (Fig. 6a) in turn influences the zonally anomalous distribution of moisture, which, as already discussed, features negative $\overline{q}^*_{850}$ values over the Mediterranean Sea (Fig. 5a). This observation clearly demonstrates how zonal asymmetries in moisture cannot simply be interpreted as a thermodynamic response to the land-sea contrast, but are strongly influenced by the stationary vertical motions. These zonal moisture anomalies are then transported by the stationary circulations that induce them, with co-variations between the two zonally anomalous fields that lead to vertical advection by the zonally anomalous flow of zonally anomalous moisture $(-\langle \overline{q}^* \nabla \cdot \overline{\mathbf{u}}^* \rangle)$ that can cancel the vertical advection of zonally symmetric moisture by the zonally anomalous circulation $(-\langle [\overline{q}] \nabla \cdot \overline{\mathbf{u}}^* \rangle)$ in regions of negative (dry) zonally anomalous moisture anomalies, such as the Mediterranean Sea. In other words, the drying effect of the zonally anomalous divergent circulation is reduced by the pure stationary term in regions of reduced moisture availability $(\overline{q}^*_{850} < 0)$. Additionally, the fact that zonally varying moisture primarily varies in the meridional direction, with gradients that have opposite sign to those of the zonally averaged moisture, suggests that the same zonally varying horizontal motions will result in opposing meridional advection throughout the Mediterranean domain, leading to horizontal advective tendencies by the pure stationary term that oppose those of the zonally asymmetric circulation acting on zonally averaged moisture.

Finally, moving our attention to the term arising from the transport of zonal variations in moisture by the zonally averaged circulation $(-\nabla \cdot \langle [\overline{\mathbf{u}}] \overline{q}^* \rangle$, Fig. 4h), we note how this term has a somewhat weaker magnitude and a more spatially varying

behavior than the other two terms. Importantly, the spatial patterns of this term cannot simply be explained in terms of the divergence of zonally anomalous moisture by the zonally averaged vertical motion, exposing the important role of horizontal advection (see Fig. 7e, f).

While these simple arguments provide insights into mechanisms that lead to the observed patterns of the three stationary eddy terms, therefore putting on strong physical ground what might otherwise appear as decomposition artifacts, the contribution by wind convergence and moisture advection to the stationary eddy MFC can be explicitly quantified using Eq. 7. Shown in Fig. 7 for all stationary terms, this further decomposition confirms the more qualitative arguments provided above. Notably, it highlights how the contribution of the advective term to the pure stationary eddy MFC is at least as, if not more important than, the wind divergence term (Fig. 7a,b).


     To summarize, our findings show how the total stationary eddy MFC arises firstly from a strongly divergent zonally anomalous circulation, which diverges zonal-mean moisture away from the Mediterranean region. This is partly compensated for by the pure stationary eddy MFC, through both wind divergence and advection of zonally anomalous moisture by the zonally anomalous circulations that partly induce these anomalies. Zonal variations in moisture also impact net precipitation in the

Mediterranean region through advection by the zonal-mean circulation, which, while more spatially varying and weaker in magnitude than the other two terms, further counterbalances moisture divergence by the zonally anomalous circulation, especially over the Sea. In other words, zonal variations of moisture primarily influence the moisture budget of the Mediterranean region through horizontal advection. It is in fact interesting to note how the contribution to the total stationary eddy MFC from the horizontal advection of zonally anomalous moisture (Fig. 8) has spatial patterns that, especially over the ocean, are grossly

similar, but of opposite sign, to the transient eddy MFC (Fig. 4d). This confirms findings from global analyses that suggest that transient eddies transport moisture down the moisture gradients set up by the time-mean horizontal advection, leading to a partial cancellation of these terms (cf. Wills and Schneider, 2015).

## 3.2    Seasonal Cycle

We analyse the seasonality of the hydroclimate in the Mediterranean region using four seasons expressed as DJF (December to February), MAM (March to May), JJA (Jun to August), and SON (September to November). Here we focus primarily on the solstice seasons, DJF and JJA, both showing zonal averages in the Mediterranean sector (Fig. 9) and spatial patterns of all relevant terms (Fig. 10 for DJF and Fig. 11 in JJA). Interested readers can find similar analyses for the shoulder MAM and SON seasons in Appendix A.

According to Fig. 9a, the dominant term in the maintenance of the sector-mean Mediterranean hydroclimate, $\{\overline{P}\} - \{\overline{E}\}$, during DJF is the convergent transient storm systems, $-\{\nabla \cdot \langle \overline{\mathbf{u}'q'} \rangle\}$. In this season, the divergent zonally-anomalous circulation term, $-\{\nabla \cdot \langle \overline{\mathbf{u}}^*[\overline{q}] \rangle\}$ is opposed by the convergent pure stationary eddies, $-\{\nabla \cdot \langle \overline{\mathbf{u}}^*\overline{q}^* \rangle\}$, and the zonally anomalous water vapor term, $-\{\nabla \cdot \langle [\overline{\mathbf{u}}]\overline{q}^* \rangle\}$. The total stationary-eddy term, $-\{\nabla \cdot \langle \overline{\mathbf{u}}^\dagger \overline{q}^\dagger \rangle\}$, therefore, has very little impact on the zonal-mean hydrological cycle during this season due to the opposing behaviours of the three contributing stationary-eddy terms.

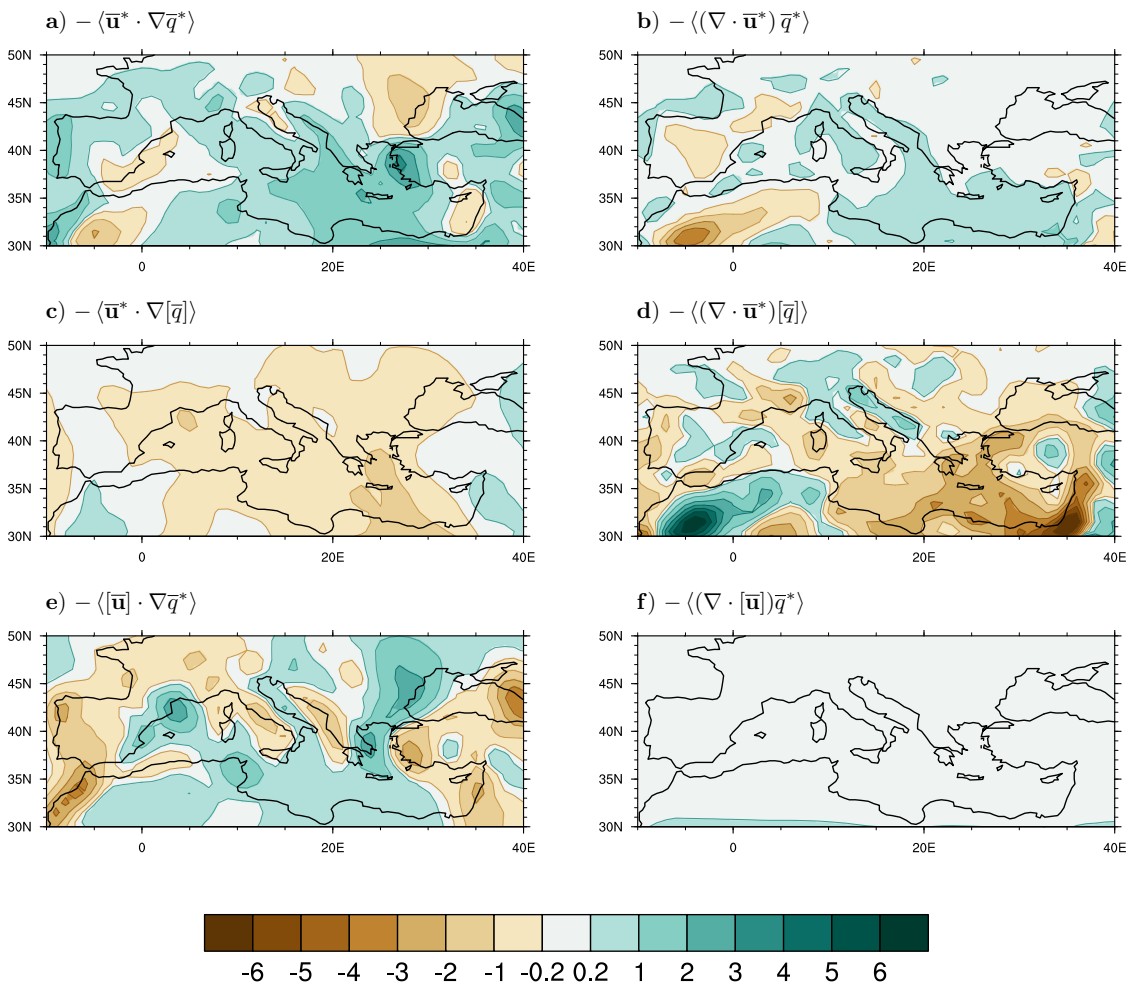

**Figure 7.** Annual mean climatological contribution from advection (left) and wind divergence (right) to the stationary eddy MFC within the Mediterranean region in the $1979-2020$ period from ERA5: (a,b) pure stationary eddies, (c,d) zonally anomalous circulation term, and (e,f) zonally anomalous water vapor term. Units are millimeters per day.

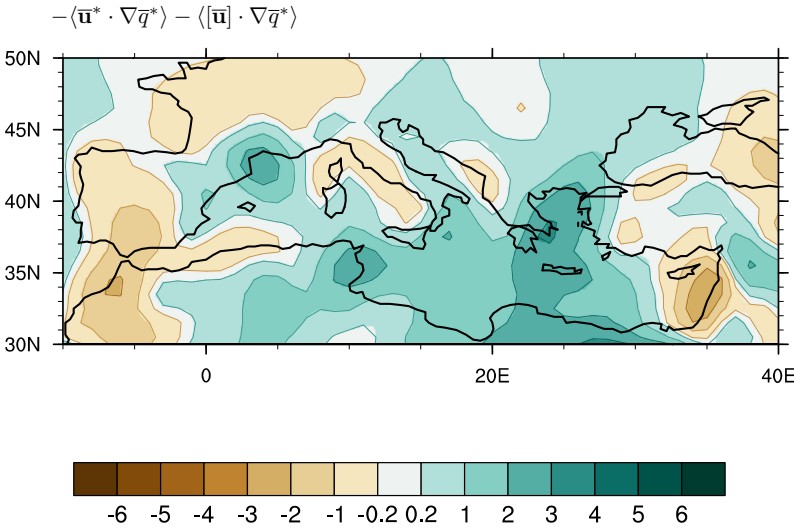

$$-\langle \overline{\mathbf{u}}^* \cdot \nabla \overline{q}^* \rangle - \langle [\overline{\mathbf{u}}] \cdot \nabla \overline{q}^* \rangle$$

**Figure 8.** Annual mean climatological contribution to $P - E$ from horizontal advection of zonally anomalous moisture (i.e., the sum of the terms shown in panels a and e of Fig. 7) from ERA5. Units are millimeters per day.

In summer (Fig. 9b), the divergent zonally anomalous circulation term dominates the budget and more than counterbalances the convergent pure stationary eddies, leading to negative $\{\overline{P}\} - \{\overline{E}\}$ in the region. Note how this term is four times as strong in JJA than in DJF (note the different x-axis ranges in Figs. 9a, 9b and A1). Interestingly, the divergent zonal-mean flow, $-\{\nabla \cdot \langle [\overline{\mathbf{u}}][\overline{q}] \rangle\}$, plays a secondary role during both DJF and JJA. Nevertheless, compared to JJA, this term causes a slightly larger drying tendency in the southern Mediterranean region during DJF.

The spatial variations of the Mediterranean moisture budget and its components in DJF and JJA are shown in Fig. 10 and Fig. 11, respectively, which clearly highlight the strong seasonal signal, by which the net flux of fresh water (i.e., $P - E$, panels a) is the most positive during DJF and the most negative during JJA. According to ERA5, and in agreement with previous work (Seager et al., 2014a), the transient eddies provide the moisture flux convergence needed to balance the positive $P - E$ over all Mediterranean land regions during winter (Fig. 10d), as the jet shifts southward. It can be seen that the Mediterranean region

experiences a strong land-sea contrast due to transient winter storms, with wet conditions over land and dry conditions over the sea itself. During summer, the signal appears more varied, with transient eddies contributing to moisture flux divergence over the Mediterranean Sea and certain land regions such as northwestern Africa, northern Italy, and southern Turkey, and contributing to moisture flux convergence in other regions such as northern land regions surrounding the sea and northern Turkey (Fig. 11d).

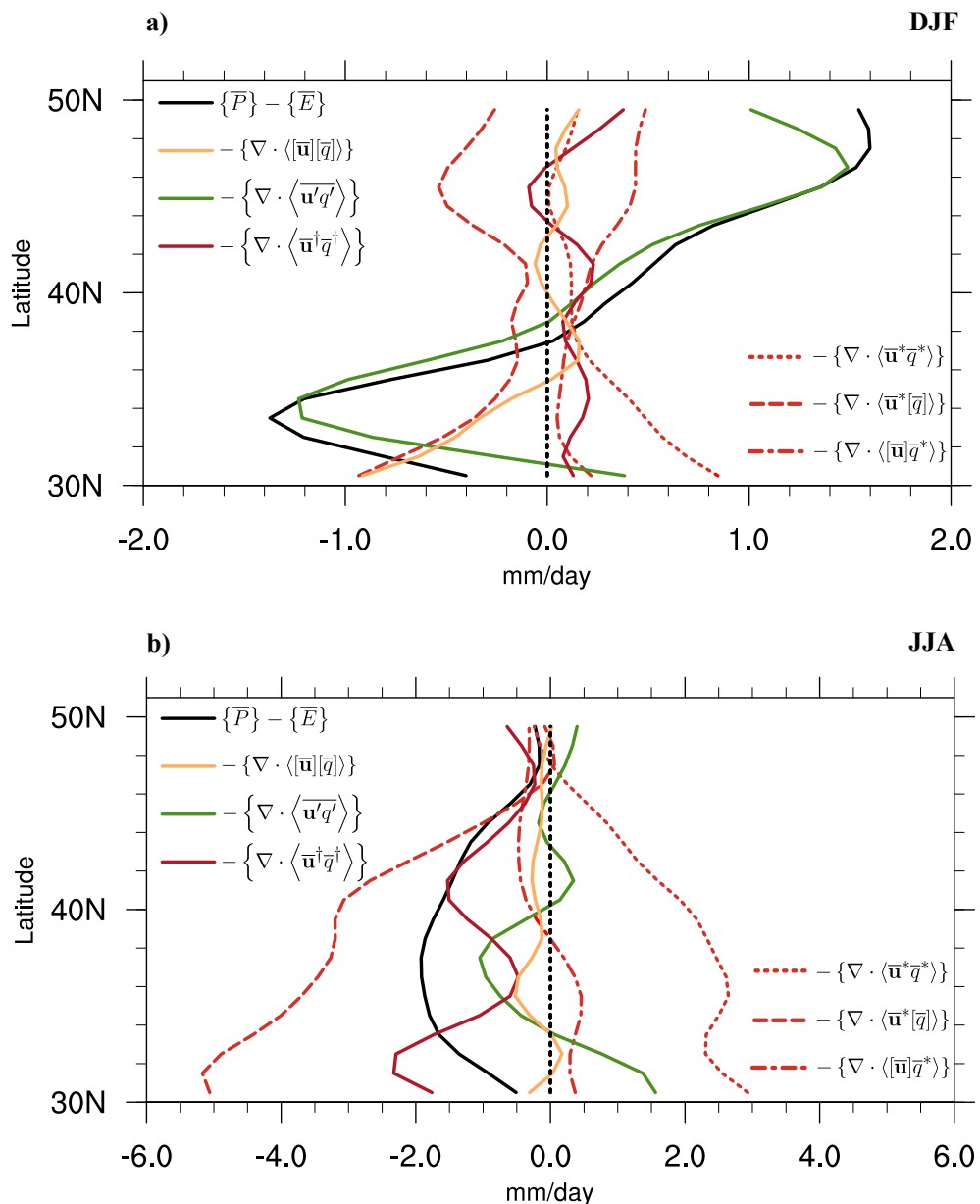

**Figure 9.** Sector-mean components of the Mediterranean moisture budget in the $1979 - 2020$ period from ERA5 during (a) DJF and (b) JJA. Each curve is smoothed in latitude using a 3-point running average filter. Note the difference in the x-axes.

The zonal-mean flow leads to subsidence and minor moisture divergence in the Mediterranean during both seasons: in DJF, it is the southern Mediterranean that undergoes mild divergence (Fig. 10c), while during JJA, it is the central and northern Mediterranean that undergo mild moisture divergence and drying (Fig. 11c).

     Finally, focusing on the contributions by stationary eddies, we see how the total stationary eddies are generally weaker during DJF than JJA, when in fact, all the terms involving stationary eddies are very pronounced (Figs. 11f, 11g, 11h). During JJA,
the total stationary eddy moisture flux causes intense drying in the Mediterranean, except in northwest Africa, the Alps, and a region extending from eastern Spain to western south France (Fig. 11e). Transports of zonal-mean moisture by the zonally anomalous circulation and by the pure stationary eddies have larger magnitudes than the term related to zonally anomalous moisture (Figs. 11f, 11g, 11h), with mostly opposing tendencies: While the zonally anomalous circulation results in strong anomalous descent linked to the monsoon-desert mechanism (Rodwell and Hoskins, 1996, 2001; Cherchi et al., 2014; Simpson
et al., 2015; Wills and Schneider, 2015; Cherchi et al., 2016) and associated low-level divergence over the central and eastern Mediterranean (not shown), except for northwest Africa, the Iberian Peninsula, and the Alps (Fig. 11g), the pure stationary eddy moisture flux mediates the resulting dry climatic conditions through moisture convergence arising from the zonally anomalous descent diverging dry anomalous moisture (not shown) over the region (Fig. 11f). Northwest Africa appears to be an exception in both cases: The moisture flux convergence by the zonally-anomalous circulation is offset by the moisture divergence due to
the pure stationary eddies.

## 4    Conclusion and Discussion

We investigated the maintenance of the atmospheric branch of the Mediterranean hydrological cycle within the ERA5 reanalysis via extended decomposition of the atmospheric moisture budget into contributions from the mean meridional circulation, stationary and transient eddies.
Starting from analysing the closure of the global moisture budget, we conclude that the global budget, averaged over land and ocean, is not closed. $P$ consistently exceeds $E$, indicating that there is an artificial source of moisture in ERA5. From the mid-1990s, the imbalance between the two fields is reduced, but this introduces unrealistic jumps in $P$ and $E$, making it difficult to utilize this data for trend analyses and hydroclimate change studies. Hence, we recommend extra caution when analysing trends in quantities related to the ERA5 moisture budget. In the Mediterranean region, the budget is also not closed,
and in fact, the annual mean value of the residual between net precipitation and moisture flux convergence is 0.136 mm per day, but also shows an improvement in the most recent decades. While a closed budget would be desirable, it does not prevent the assessment of how different circulations help maintain the net precipitation patterns, which is the goal of the analyses presented here.

     Our findings on the maintenance of the hydrological cycle in the Mediterranean region confirm previous work (Seager et al.,
2014a) that has shown how, in the long-term annual mean, the transient eddies are the dominant pathway by which moisture originating from the Mediterranean Sea converges over all surrounding land regions, except northwestern Africa. The moisture flux divergence due to the transient eddies is opposed by the moisture flux divergence due to the time-mean flow.

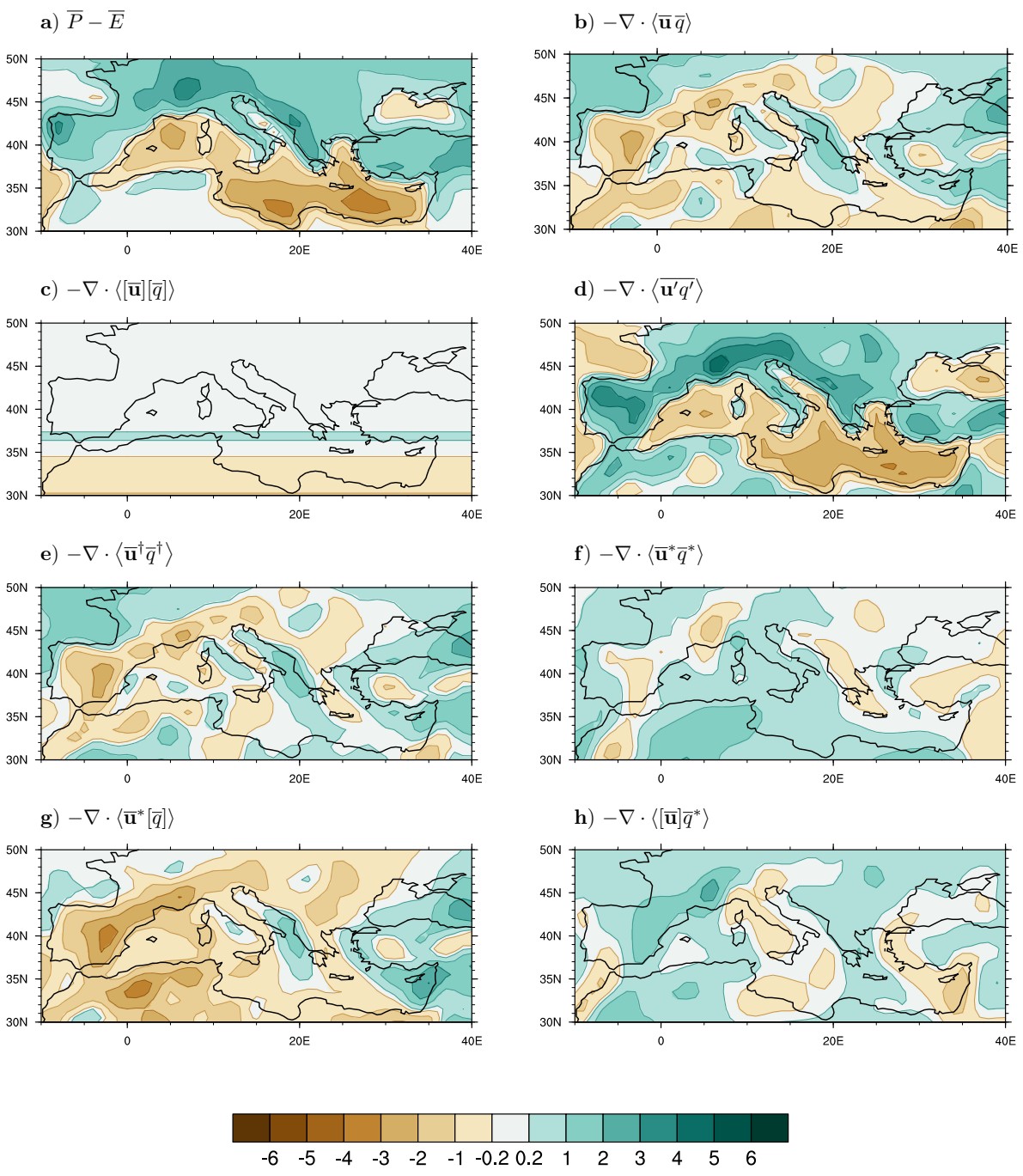

**Figure 10.** December-February (DJF) climatological Mediterranean moisture budget in the $1979 - 2020$ period from ERA5: (a) $P - E$, (b) time mean flow, (c) zonal-mean circulation, and (d) transient eddies. (e) Total stationary eddies and its components due to (f) pure stationary eddies, (g) zonally anomalous circulation, and (h) zonally anomalous moisture. Units are millimeters per day.

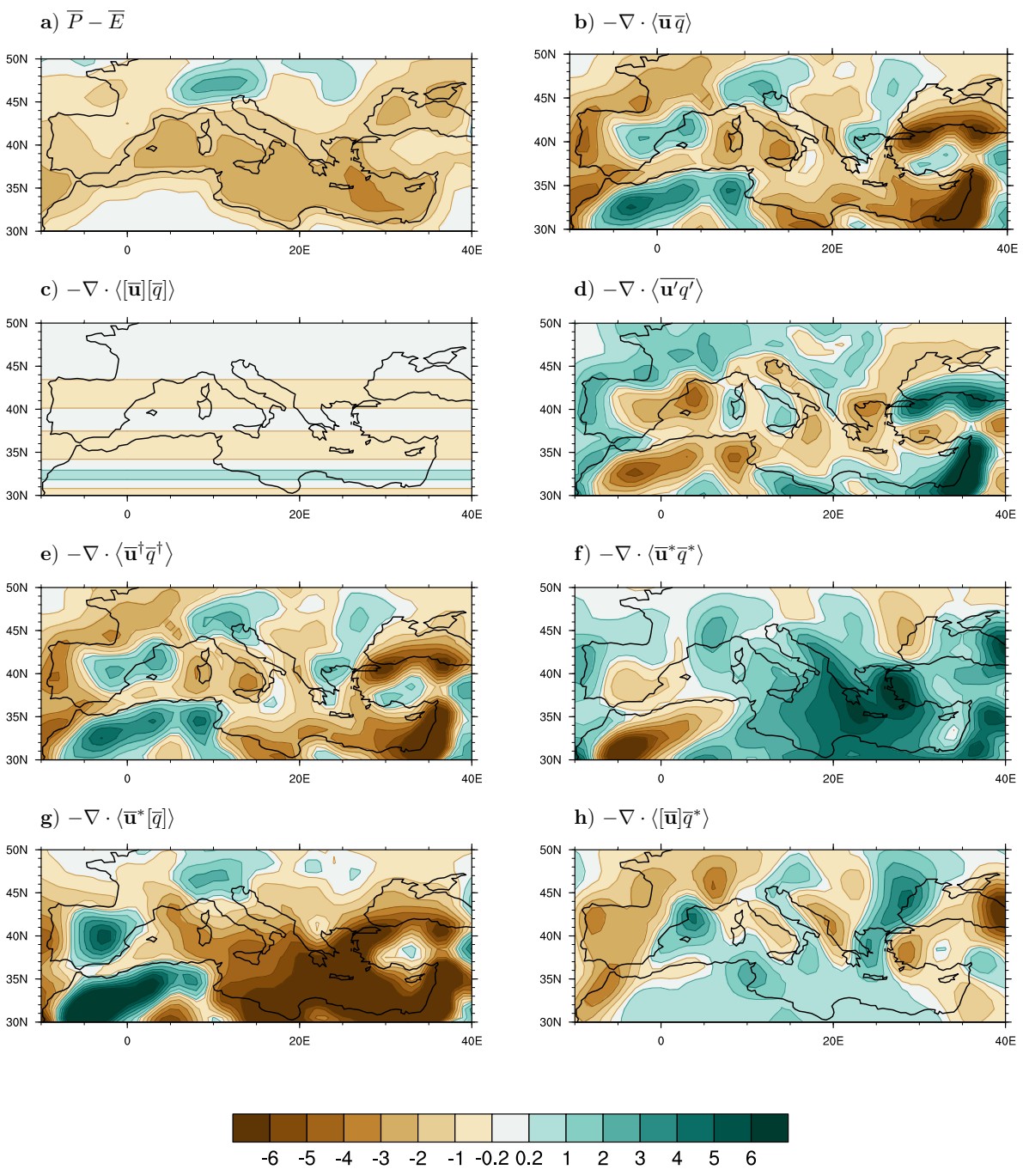

**Figure 11.** June-August (JJA) climatological Mediterranean moisture budget in the $1979 - 2020$ period from ERA5: (a) $P - E$, (b) time mean flow, (c) zonal-mean circulation, and (d) transient eddies. (e) Total stationary eddies and its components due to (f) pure stationary eddies, (g) zonally anomalous circulation, and (h) zonally anomalous moisture. Units are millimeters per day.

We further decompose the time-mean flow into contributions from the zonal-mean flow and variations about the zonal mean. According to ERA5, the long-term, annually averaged mean meridional circulation causes mild subsidence and moisture divergence in the southern Mediterranean, but is negligible everywhere else in the region. This highlights how the time-mean divergence primarily arises from zonally varying circulations and moisture patterns, and their covariation. Among all stationary eddy terms, the divergence of zonally symmetric moisture by the zonally anomalous circulation, featuring subsidence and low-level horizontal wind divergence over the central and eastern Mediterranean, appears the most dominant. Zonally anomalous moisture patterns in the region arise primarily as a dynamical response to these zonally anomalous vertical motions rather than just as a thermodynamic response to the land-sea contrast. As a consequence, the pure stationary eddy term, through both divergence and advection of the zonally anomalous moisture by the zonally anomalous circulations that induce these anomalies, partially compensates the divergence of zonally symmetric moisture by the zonally anomalous circulation. Zonal variations in moisture also impact net precipitation in the Mediterranean region through advection by the zonal mean circulation, which, while weaker in magnitude than the other two terms, further counterbalances moisture divergence by the zonally anomalous circulation, especially over the Sea. In other words, net precipitation patterns over the Mediterranean region are primarily dominated by the zonally varying circulation, with zonal moisture anomalies playing a more minor role.

In winter, positive net precipitation over land regions is primarily sustained by transient storms, which converge moisture originating from the Mediterranean Sea over the surrounding land masses. Stationary eddies are weaker in this season and are not large enough to counteract the influence of the transient eddies. In summer, the strengthened moisture flux divergence due to the stationary eddies dominates the hydrological cycle. This leads to net evaporation over both sea and land in the Mediterranean region, with the exception of the Alps. While pure stationary eddies provide moisture convergence, they are opposed and offset by the divergence of zonally symmetric moisture by the zonally anomalous circulation through mechanisms similar to those discussed for the annual mean.

By showing some of the essential features of the Mediterranean hydrological cycle as well as the role of different circulations on the maintenance of $P - E$ within the region in the annual, seasonal, and zonal means, our findings might help shed further light on future changes in the Mediterranean and the resulting impacts on water resources. In particular, by exposing the explicit role of zonally anomalous circulations, they add one additional piece to the complex puzzle of the hydroclimate change in the region. Expansion of the subtropical dry zones in association with the projected expansion of the Hadley cell and the midlatitude jet (e.g., Lu et al., 2007; Lucas et al., 2014; Grise and Davis, 2020) is often invoked as implicated in the projected increased aridification in the Mediterranean under greenhouse gas forcing, even if mechanisms remain debated (e.g., Staten et al., 2020). The importance of zonally anomalous descent and lower-level diverging wind patterns for the maintenance of the climatological net precipitation patterns emerging from our study, however, suggests that the strength of the response cannot simply be explained in terms of zonally invariant processes and that dynamical changes in stationary eddies might be among the factors making the Mediterranean region a "hot and dry" spot for global climate change.

It is important to note that recent work (e.g., Li et al., 2022; Galanti et al., 2022) has emphasized how the meridional overturning circulation, traditionally interpreted as zonally uniform, has in fact a significant longitudinal structure with profound impacts on regional climate. This has led to the emergence of a regional Hadley cell perspective, with the introduction of a

local meridional overturning, similar to the global one, but with a longitudinal distribution based on the irrotational flow of the atmospheric circulation (e.g., Zhang and Wang, 2013; Nguyen et al., 2018; Li et al., 2022; Galanti et al., 2022; Raiter et al., 2024). Preliminary analyses we have conducted using this framework, however, suggest that the descending motion associated with the regional meridional overturning is weak over most of the Mediterranean and explains at most between half to two thirds of the observed descent, even in regions, such as Egypt and Libya, where it is the strongest. In other words, even when accounting for its longitudinal variations, the meridional overturning circulation does not capture all zonal asymmetries of the mean flow and their influence on the net precipitation of the Mediterranean region, and stationary waves need to be considered. More quantitative assessments are left for future work. Also left for future work is the study of how the dynamics exposed in this study is well captured and projected to change by state-of-the-art climate models.

## Appendix A

Contributions from each term to the sector-mean Mediterranean moisture budget in the $1979 - 2020$ period during spring (MAM) and autumn (SON) are shown in Fig. A1. During both MAM and SON, the region features positive $\{\overline{P}\} - \{\overline{E}\}$ in its northern part and negative $\{\overline{P}\} - \{\overline{E}\}$ in its southern part (Fig. A1a, A1b). As shown in section 3.2, the same pattern is also evident for the zonal-mean Mediterranean hydroclimate during DJF (Fig. 9a), in contrast to the widespread negative $\{\overline{P}\} - \{\overline{E}\}$ over all latitudes within the region in summer.

During spring, the stationary-eddy moisture flux divergence arises mainly from the divergence of zonal-mean moisture by the zonally anomalous circulation, which is partially opposed by pure stationary eddy moisture convergence (Fig. A1a). While having little impact on the zonal-mean hydroclimate of the region, during spring the mean meridional circulation causes moisture flux divergence (convergence) south (north) of $\sim36°$N. During autumn, instead, the mean meridional circulation causes strong subsidence and divergence at all latitudes within the region (Fig. A1b). In both seasons, the overall $P - E$ patterns are primarily explained by the transient eddies, which converge to higher latitudes moisture originating from lower latitudes.

The spatial patterns of all terms for MAM and SON are shown in Fig. A2 and Fig. A3, respectively. During spring, the northern Mediterranean land regions, such as the Alps and the Balkans, experience an excess of $P$ over $E$ (Fig. A2a). According to ERA5, the mean meridional circulation has its weakest contribution to the region's hydroclimate in this season (Fig. A2c). As seen in winter (see Fig. 10), overall transient eddies diverge moisture from the sea and converge it over the surrounding land regions, with the exception of northwestern Africa (Fig. A2d). The total stationary eddies tend to reinforce the transient-eddy tendency over the sea, but oppose it over land (Fig. A2e), with the largest contribution arising from the zonally anomalous circulation term (Fig. A2g). Over the Mediterranean Sea, the transports of zonally anomalous moisture by the zonally symmetric flow and by pure stationary eddies cause convergence and wet conditions, which partially oppose the divergence due to the zonally anomalous circulation (Fig. A2f, A2h).

During autumn, all northern Mediterranean land regions have strong excess of $P$ over $E$ (Fig. A3a). The moisture divergence by the mean meridional circulation causes a widespread drying tendency over the central and southern parts of the region, and

is at its strongest (Fig. A3c). The total stationary eddy term is still primarily explained by the zonally anomalous circulation term, but unlike all other seasons, this term causes strong divergence only over the Eastern Mediterranean and in fact results in moisture convergence over Italy and the Adriatic and the Tyrrhenian Seas (Fig. A3e). Over the eastern Mediterranean, the divergence of mean moisture by the zonally anomalous circulation term is opposed by moisture convergence by the pure

stationary-eddy and the zonally anomalous moisture components (Fig. A3f, A3h).

*Code and data availability.*  ERA5 data are available from the Copernicus Climate Data Store (CDS). Codes developed for the analyses and resulting data are available upon request.

*Author contributions.*  SB conceived this study, RT performed the analyses, RDA provided some of the codes, and all authors discussed the results. RT and SB wrote the manuscript with some helpful feedback from RDA.

*Competing interests.*  The authors declare they do not have any competing interests.

*Acknowledgements.*  We thank Robert C. J. Wills, Annalisa Cherchi, two anonymous reviewers, and the editor, Nili Harnik, for their helpful comments on drafts of this manuscript. All authors acknowledge funding from the European Union under Next Generation EU, Mission 4 Component 2 - CUP E53D23021930001. SB also acknowledges support from the European Union - Next Generation EU, Mission 4 Component 2 - CUP E63C22000970007.

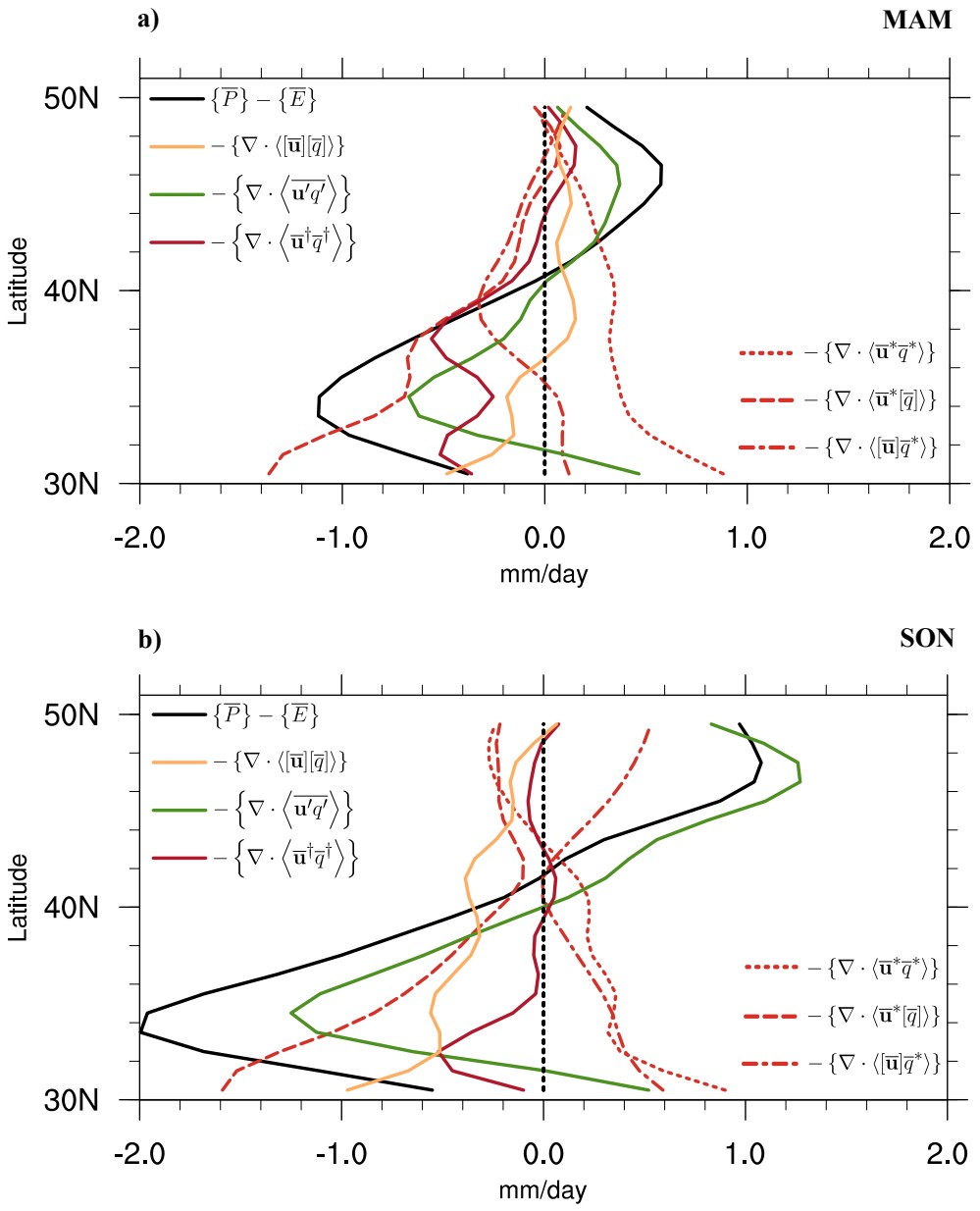

**Figure A1.** Sector-mean components of the Mediterranean moisture budget in the $1979 - 2020$ period from ERA5 during (a) MAM and (b) SON. Each curve is smoothed in latitude using a 3-point running average filter.

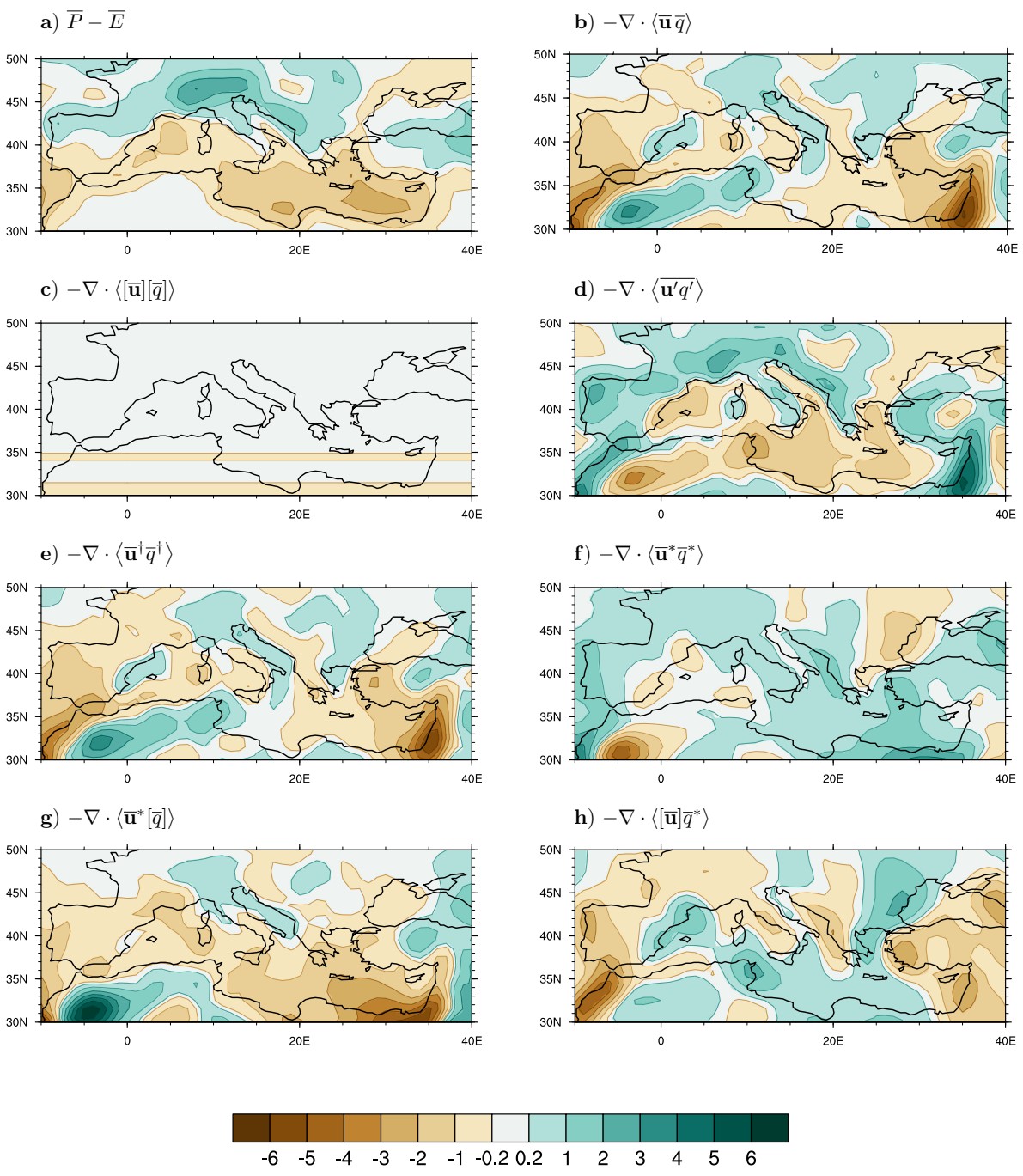

**Figure A2.** March–May (MAM) climatological Mediterranean moisture budget in the $1979 - 2020$ period from ERA5: (a) $P - E$, (b) time mean flow, (c) zonal-mean circulation, and (d) transient eddies. (e) Total stationary eddies and its components due to (f) pure stationary eddies, (g) zonally anomalous circulation, and (h) zonally anomalous moisture. Units are millimeters per day.

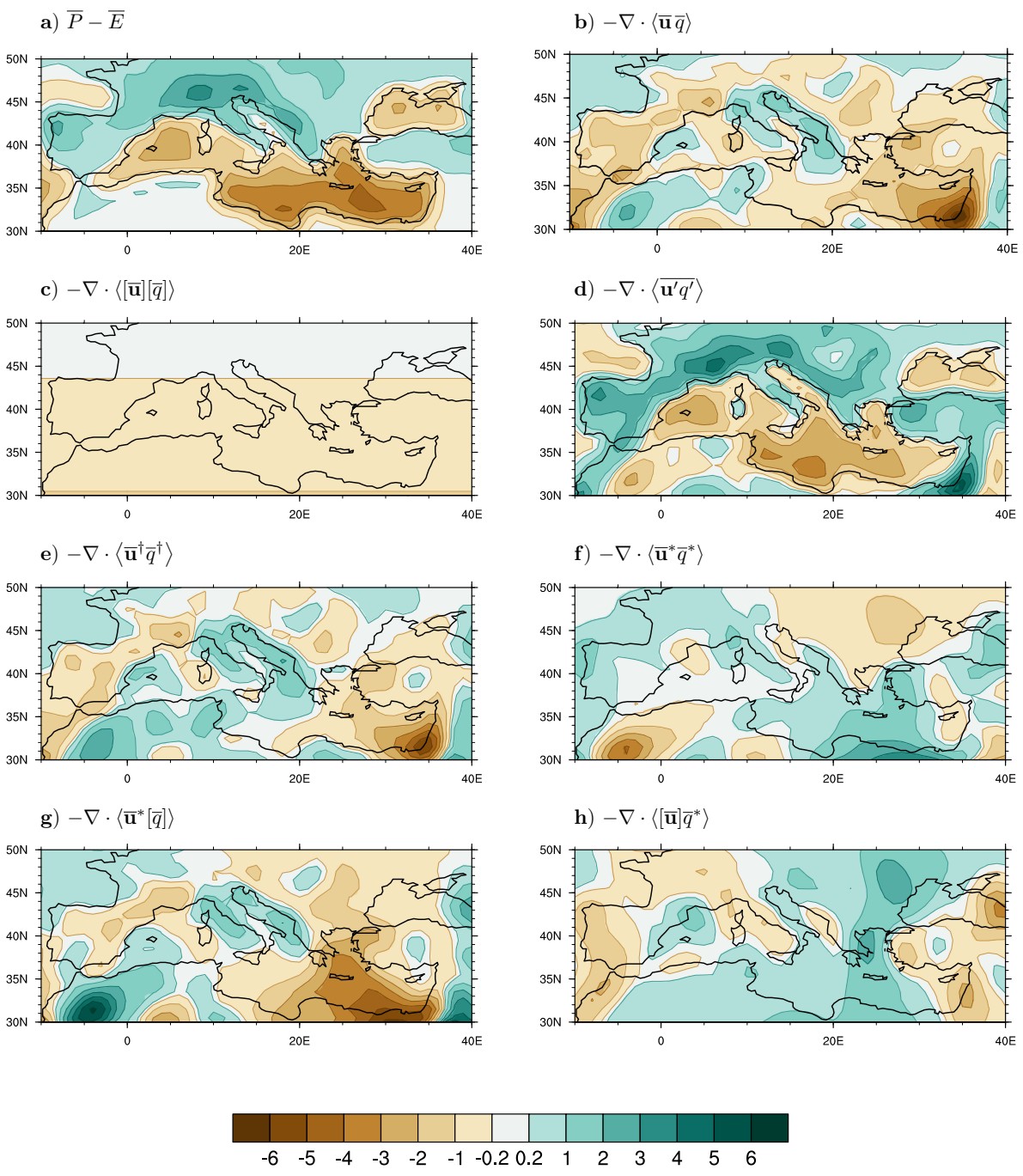

**Figure A3.** September-November (SON) climatological Mediterranean moisture budget in the $1979 - 2020$ period from ERA5: (a) $P - E$, and (b) time mean flow, (c) zonal-mean circulation, (d) transient eddies. (e) Total stationary eddies and its components due to (f) pure stationary eddies, (g) zonally anomalous circulation, and (h) zonally anomalous moisture. Units are millimeters per day.

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
