# Peer review of "Revisiting the Moisture Budget of the Mediterranean Region in the ERA5 Reanalysis"

_EGUsphere, 2024_

## Referee Comment (RC2)

Review of

**Revisiting the Moisture Budget of the Mediterranean Region in the ERA5 Reanalysis**

Tootoonchi et al.

**General**

The authors analyze the moisture budget of the Mediterranean region using the ERA5 reanalysis. Their key findings are that moisture budget is not sufficiently closed to allow detection of trends, and the critical contributions of zonally anomalous terms. The writing is generally clear and the analysis sufficiently technically proficient. However, I find the work lacking in several regards. Primarily, the analysis for the most part does do much in terms of relating the various terms (e.g., stationary and transient eddies) to relevant physical processes (e.g., relating stationary eddies to rainy storms associated with the subtropical jet during winter). The authors also do not do a good job of delineating their findings from previous work. Nevertheless, the analysis can potentially help shed light on key processes in the Mediterranean hydrological cycle. I therefore recommend accepting the paper after addressing my major comments, provided below.

**Major comments**

1.  The stated objective of this analysis, to better understand the contributions of time mean and stationary eddies (lines 58–59), is rather incremental. On top of that, the authors do little to delineate their results from previous work. It is therefore not clear what are actual novel findings of the analysis. Further, in several places, it is stated that the results are consistent with previous works, without citing those works. Aside for being bad practice, this further obfuscates the potential novel contributions of the present analysis. In summary, the authors should do a much better job of referencing previous work and clearly stating the novel findings of the present analysis.

2.  It is interesting to note that the periods during which global mean $P$ does not equal $E$ go along with rapid global warming (Figure 1). $P$ does not equal $E$ during the rapid global warming from 1979 to the end of the previous century, ending in the 97/8 Niño event. This is followed by a period of weak temperature increase (the so-called 'global warming hiatus') when the $P - E$ residual is small. Then, $P$ deviates from $E$ again as the rate of global warming increases again after 2012. Can you convince the reader that the global residual of $P - E$ is not due to the moisture storage term in the moisture equation $\langle \dot{q} \rangle$ or due to inaccuracies in your methodology? Specifically, one

can use Clausius Clapeyron (CC) to demonstrate the former. Under constant relative humidity $H$, we would get $\langle \dot{q} \rangle \approx H \langle q \rangle \alpha \dot{T}$ where $\alpha$ is the CC parameter (~7%) and $\dot{T}$ is the rate of global warming. Given that the global mean of precipitable water is about 20mm, this yields for a rate of global warming between 1979−2000 of 0.03K/year $\langle \dot{q} \rangle \approx 10^{-4} mm/day$, justifying the assumption of steady state. It therefore remains to make the case that the residuals are not due to your methodology. For example, due to the use of fewer than available vertical levels, or omitting from the integral near-surface values in regions where surface pressure exceeds 1000hPa. (One way of estimating the integration error may be to compare precipitable water values provided in ERA5 with those derived by integration.) Another potential source of the residual is changes in ice volume. In summary, the authors should convince the reader that the residual of $P - E$ is indeed a feature of the ERA5 reanalysis and not due to their methodology.

3. Given that one potential novelty of this work is to demonstrate the contribution of zonal anomalies in either the humidity or the winds, showing only the immediate Mediterranean region make it difficult to see whether zonally asymmetric terms are related to zonal overturning circulation with links to either the Atlantic or to Asia and Arabia. For example, what is the driver of the negative contribution in the eastern Mediterranean by the dynamic term shown in Fig. 8g? Could this be related to the descending branch of the Indian monsoon (i.e., the so-called Monsoon-Desert mechanism by Rodwell and Hoskins)? Expounding on such processes in the analysis and zonally extending the analysis region may shed light on such potential drivers of the various patterns (e.g., the Indian monsoon, the Persian trough, ventilation of land areas, etc.), which are hardly discussed in the text.

**Comments by line number**

10−13      This sentence is cryptic

48            I would also add Elbaum et al. (2022, "Uncertainty in projected changes in precipitation minus evaporation: Dominant role of dynamic circulation changes and weak role for thermodynamic changes.")

122          Why is only a subset of the vertical levels used for the vertical integration? Wouldn't this reduce the accuracy of the vertical integration?

185−186    This gives the impression that, as in the global mean, there is some constraint under which we would expect $P - E$ to vanish when averaged regionally, which is not the case.

213 and elsewhere      What previous work? Please specifically cite relevant works.

245—247    Not sure I agree with this statement. The terms $\nabla \cdot \langle \bar{u}^* \bar{q}^* \rangle$ and $\nabla \cdot \langle \bar{u}^* [\bar{q}] \rangle$ generally balance out, and so the term $\nabla \cdot \langle [\bar{u}] \bar{q}^* \rangle$ would seem to be of the same order as their residual. More generally, the term $\nabla \cdot \langle \bar{u}^* [\bar{q}] \rangle$ would be related to the zonally asymmetric circulation, whereas the $\nabla \cdot \langle [\bar{u}] \bar{q}^* \rangle$ term would be related to zonally asymmetric temperature variation. If indeed the latter term is not significant, how does this sit with the alleged immense importance of land-ocean contrasts?

255         Why do the transient eddies dominate the sector mean? Please provide an explanation backed by the relevant references. Jet?

Figure 4: the mean and stationary terms in Fig. 4e and 4h are nearly identical. Can you explain why? Please comment on this.

317—321    Note that recently, Adam et al. (2023, "Reduced Tropical Climate Land Area Under Global Warming.") showed that over land areas the subtropics are expanding on both their poleward and equatorward edges, and that this expansion is likely driven by thermodynamic drying (reduced evaporative cooling), rather than a dynamic expansion of the tropical overturning circulation.

---

## Author Response (AR1)

**Response to reviewers' comments on the manuscript "Revisiting the Moisture Budget of the Mediterranean Region in the ERA5 Reanalysis"** - Roshanak Tootoonchi; Simona Bordoni; Roberta D'Agostino

We thank both reviewers for their careful reading and their constructive comments on the manuscript, which helped improve its quality. We addressed all reviewers' suggestions in our revised submission with the following major changes:

- We modified the colour schemes in Figures 3, 7 and A1 in order to allow readers with colour vision deficiencies to interpret them correctly. We reproduce the modified version of Fig. 3 of the manuscript here as an example (Fig. R1).

- We have significantly expanded the results discussion provided in Section 3 and performed additional analyses with the addition of several new figures to put our overall results on better physical ground. In particular, we now explain why the stationary terms in the moisture budget decomposition are what they are and discuss possible physical reasons behind partial cancellation between the different terms.

- Wherever possible, we modified the text to highlight the novelty of our work more elaborately and to reference previous work appropriately.

A detailed, point-by-point response (in black) to the reviewers' comments (in blue) is provided below.

**Reviewer 1**

We thank Reviewer 1 for taking the time to provide such detailed review of the manuscript, touching all aspects of the paper and mentioning the key findings.

This paper examines the climatological moisture budget - the atmospheric branch of the hydrological cycle - in the latest ERA5 atmospheric reanalysis for the annual mean and by season. The authors first rather usefully point out the global imbalance between P and E in ERA5 with P exceeding E and that the imbalance has reduced over time as E has increased substantially - primarily through an abrupt increase in the mid 1990s. I take this to mean that there is an artificial moisture "source" in ERA5 - though the authors say "sink" - which allows global mean P to exceed what is supplied by E. The authors also usefully show the moisture imbalance between P-E and the convergence of the vertically integrated moisture budget for the Mediterranean region. It is not small either, though it has been reducing in size over time, though not with the same time evolution as the global imbalance. Because of these matters, the authors caution on the analysis of trends in the ERA5 moisture budget, which is sage counsel. The authors then move on their decomposition of the climatological moisture budget. Their analysis with a new version of ECMWF reanalysis largely confirms prior findings of Seager et al. (2014) with ERA-Interim. The transient eddies are found to extract moisture from the Mediterranean Sea and converge it

[Figure]

Figure R1: Sector-mean annually averaged (ANN) components of the Mediterranean moisture budget in the $1979 - 2020$ period from ERA5. [Color-scheme corrected for readers with color deficiencies]

over land everywhere. The authors compute this term as a residual form monthly data but the pattern agrees with Seager et al. who commuted this directly from daily data, which would be worth pointing out. Unlike the prior work, the authors then decompose the mean flow moisture component into zonal mean and stationary wave components and then these into terms due to zonal asymmetries of humidity and mean circulation. This has not been shown before. To my surprise the component due to the zonal mean (Hadley Cell) overturning is very small despite this term oft being quoted as the reason for the semi-aridity of the Med region. This is rue even in winter. The work emphasizes the importance of the stationary waves and finds drying is primarily provided by the zonally asymmetric circulation (not humidity). My main criticism of the paper is that why the contributions from the stationary eddies are what they are is not explained. The authors should seek to explain, in terms of the zonal asymmetries, why the the transport of zonally symmetric moisture by the zonally anomalous circulation is a drying in the Med while the transport of zonally anomalous moisture by the zonally symmetric circulation is a wetting and, also, why the put stationary eddy term is mostly a convergence. Currently these are presented as decomposition artifacts but instead need to be put on good physical ground.

We thank the reviewer for this comment, which echoes similar remarks by Reviewer 2. We do acknowledge that in the submitted manuscript, the contributions from the stationary eddies were presented as decomposition artifacts and not linked in a physical sense to zonal asymmetries in circulation and moisture. In doing so, we also failed to more explicitly connect our results to already published work. To address this concern, we have significantly extended our analysis and added the following discussion and related figures to Section 3.

[Figure]

Figure R2: Annual mean climatological low level (850 hPa) a) zonally anomalous moisture and b) zonal mean moisture from ERA5 in the $1979 - 2020$ period. Units are $g\,kg^{-1}$.

A good starting point to understand why the different stationary terms contribute the way they do to net precipitation in the Mediterranean region is to examine patterns of zonally asymmetric moisture and circulation individually. Considering that moisture is primarily concentrated at low levels, for the former we choose 850 hPa as a representative level, and in Fig. R2 we show both the zonal mean and variations from the zonal mean of specific humidity at this level ($[\overline{q}_{850}]$ and $\overline{q}^*_{850}$, respectively). Low-level moisture over the southern Mediterranean region and the Mediterranean Sea itself is smaller than the global zonal mean (Fig. R2a), resulting in negative (dry) zonal anomalies over these areas. We do not see a clear land-sea contrast in these anomalous humidity patterns; rather, the signal is one of a meridional contrast between a northern moister ($\overline{q}^*_{850} > 0$) part and a southern drier ($\overline{q}^*_{850} < 0$) part. An exception to this overall behavior is northwestern Africa, where $\overline{q}^*_{850}$ is positive. Not particularly informative, Fig. R2b shows the expected pattern of decreasing zonal mean specific humidity ($[\overline{q}_{850}]$) with latitude.

Characterizing zonally asymmetric circulation patterns that influence net precipitation is less straightforward, as stationary eddies can impact moisture flux convergence through both horizontal advection and vertical motion (horizontal divergence). Given the well-established importance of vertical motion on large-scale zonally anomalous $P - E$ patterns (e.g., Wills and Schneider, 2015), in Fig. R3 we show 500-hPa zonally anomalous vertical pressure velocity $\overline{\omega}^*_{500}$, which well correlates with low-level divergence and is not affected by numerical noise. Please, note that in Fig. R3 we show the negative vertical pressure velocity, so that positive (negative) values indicate ascending (descending) motion.

At 500 hPa, the zonally anomalous vertical velocity, $\overline{\omega}^*$, results in strong descent across the Mediterranean region, except northwestern Africa, where the Atlas mountain range is located, and the northern Balkans (Fig.R3a). Anomalous ascending motion can be seen in the Middle East over the Zagros mountains, with enhanced subsidence to both their east and west, the latter extending over the central and eastern Mediterranean. Notice how subsidence maximizes over the eastern

[Figure]

Figure R3: Annual mean climatological mid-tropospheric (500 hPa) a) anomalous vertical velocity and b) zonal-mean vertical velocity from ERA5 in the $1979 - 2020$ period. Units are $Pa\,s^{-1}$. Note how here we show negateve pressure velocity, hence positive values (green) denote ascending motion and negative values (brown) denote descending motion.

Mediterranean region. This is in strong agreement with previous published papers, which link the subsidence over the Mediterranean region to the influence of topography, more specifically the Zagros mountains (Simpson et al., 2015) and the Atlas mountains (Rodwell and Hoskins, 1996), and its modulation of subsidence induced by the Indian monsoon heating (Cherchi et al., 2014, 2016; Rodwell and Hoskins, 1996; Simpson et al., 2015). The global zonal mean vertical velocity, $[\overline{\omega}]$, instead, reveals a region of negligible ascending motion between 40°N and 50°N, flanked by descent to its south and weak ascent to its north (Fig.R3b).

To the extent that horizontal advection is negligible, the moisture flux convergence associated with the three stationary terms, at least in terms of overall spatial patterns, can be understood considering how the zonally averaged and/or the zonally anomalous vertical motion act on the zonally averaged and/or the zonally anomalous moisture (see also Wills and Schneider, 2015). In addition to spatial patterns, we are also interested in explaining the partial cancellation between contributions arising from the transport of zonal mean moisture by the zonally anomalous circulation ($-\nabla \cdot \langle \overline{\mathbf{u}}^*[\overline{q}] \rangle$) and the pure stationary eddies ($-\nabla \cdot \langle \overline{\mathbf{u}}^*\overline{q}^* \rangle$).

A comparison of Fig. 5b in the manuscript and Fig. R3a below shows how patterns of convergence of zonal mean moisture by the zonally anomalous circulation, $-\nabla\cdot\langle \overline{\mathbf{u}}^*[\overline{q}] \rangle$, are well explained by patterns of the zonally anomalous mid-tropospheric vertical velocity $\overline{\omega}^*_{500}$, with moisture divergence (convergence) coinciding with regions of subsiding (ascending) motion. In other words, the leading-order effect in this term is due to zonal variations in vertical motion, and associated horizontal wind convergence, which is primarily descending over the Mediterranean and diverging moisture away from the region.

The zonally anomalous descending motion in turn influences the zonally anomalous distribution

of moisture, which, as already discussed, features negative $\overline{q}^*_{850}$ values over the Mediterranean Sea (Fig. R2). This observation clearly demonstrates how zonal asymmetries in moisture cannot simply be interpreted as a thermodynamic response to the land-sea contrast, but are strongly influenced by the stationary vertical motions.

These zonal moisture anomalies are then transported by the stationary circulations that induce them, with co-variations between the two zonally anomalous terms that lead to moisture convergence/divergence by the pure stationary eddy term ($-\nabla \cdot \langle \overline{\mathbf{u}}^* \overline{q}^* \rangle$) that partially cancels that due to the transport of zonally symmetric moisture by the zonally anomalous circulation ($-\nabla \cdot \langle \overline{\mathbf{u}}^* [\overline{q}] \rangle$). In particular, over the Mediterranean region, pure stationary moisture flux convergence arises from zonally anomalous descent diverging negative (dry) moisture anomalies.

Finally, moving our attention to the term arising from the transport of zonal variations in moisture by the zonally averaged circulation ($-\nabla \cdot \langle [\overline{\mathbf{u}}] \overline{q}^* \rangle$), we note how this term has a somewhat weaker magnitude and a more spatially varying behavior than the other two terms. Importantly, the spatial patterns of this term cannot simply be explained in terms of the divergence of zonally anomalous moisture by the zonally averaged vertical motion, exposing the important role of horizontal advection.

While these simple arguments provide insights into mechanisms that lead to the observed patterns of the three stationary terms, therefore putting on strong physical grounds what might otherwise appear as decomposition artifacts, we can more rigorously decompose the moisture flux convergence terms into contributions by wind convergence and moisture advection. For instance, taking the pure stationary eddy term as an example, $-\nabla \cdot \langle \overline{\mathbf{u}}^* \overline{q}^* \rangle = -\langle \overline{q}^* \nabla \cdot \overline{\mathbf{u}}^* \rangle - \langle \overline{\mathbf{u}}^* \cdot \nabla \overline{q}^* \rangle$[1]. This decomposition, shown in Fig. R4 for all stationary terms, confirms the more qualitative arguments provided above. The one exception worth highlighting is how the contribution of the advective term to the pure stationary eddy moisture flux convergence is at least as, if not more important than, the wind divergence term (Fig. R4a,b).

To summarize, our findings show how the total stationary-eddy moisture flux arises firstly from a strongly divergent zonally anomalous circulation, which diverges zonal mean moisture away from the region. This is partly compensated for by the pure stationary eddy moisture flux convergence, through both divergence and advection of zonally anomalous moisture by the zonally anomalous circulations that induce these anomalies. Zonal variations in moisture also impact net precipitation in the Mediterranean region through advection by the zonal mean circulation, which, while more spatially varying and weaker in magnitude than the other two terms, further counterbalances moisture divergence by the zonally anomalous circulation, especially over the Sea. In other words, zonal variations of moisture primarily influence the moisture budget of the Mediterranean region through horizontal advection. It is in fact interesting to note how the contribution to the total stationary moisture flux convergence from the horizontal advection of zonally anomalous moisture (Fig. R5), has spatial patterns that, especially over the ocean, are grossly similar, but of opposite sign, to the transient eddy moisture flux convergence. This confirms findings from global analyses that suggest that transient eddies transport moisture down moisture gradients set up by the time-
* * *
[1]By taking the derivatives inside the vertical integral to compute the advective and divergent components of the moisture flux convergence, a surface term will appear. This term, which we estimate as the residual between the total stationary moisture flux convergence and the sum of all divergent and advective terms, is found to be negligible compared to all other terms.

mean horizontal advection, leading to a partial cancellation of these terms (cf Wills and Schneider, 2015).

While this discussion focuses on the annual average, similar considerations hold true for the seasonal cycle as well, which we state explicitly in Section 4. We hope that these additional analyses will address the reviewers' concerns.

I also wonder if the authors have thought about these decomposition into mean and zonally varying in terms of a zonally varying Hadley Cell as in Li et al. (2022, JGT-Atmospheres, 10.1029/2022JD036940). That would allow growing the zonal symmetric part with the part due to a zonally varying meridional overturning into one "Hadley Cell" term. I am not sure if this would capture all of the term due to zonal asymmetries of the mean flow shown here but it would be interesting to see.

Thanks for suggesting this paper. We still have to think more deeply about the advantages provided by a 'regional' view of the Hadley cell. While it is true that the zonally averaged Hadley cell is a somewhat artificial construct that does not have any bearing to regional patterns, one of its great advantages is that it is amenable to the development of simple(r) theories based on zonally averaged budgets (such as the angular momentum and energy budgets). The traditional decomposition we use here also allows for easier comparison with already published work. We will, however, keep this article in mind for future work.

Line 260 what term does "this term" refer to?

The sentence is revised in the manuscript to: "Note how the zonally anomalous circulation term is four times as strong in JJA than in DJF..."

**Reviewer 2**

**General** The authors analyze the moisture budget of the Mediterranean region using the ERA5 reanalysis. Their key findings are that moisture budget is not sufficiently closed to allow detection of trends, and the critical contributions of zonally anomalous terms. The writing is generally clear and the analysis sufficiently technically proficient. However, I find the work lacking in several regards. Primarily, the analysis for the most part does do much in terms of relating the various terms (e.g., stationary and transient eddies) to relevant physical processes (e.g., relating stationary eddies to rainy storms associated with the subtropical jet during winter). The authors also do not do a good job of delineating their findings from previous work. Nevertheless, the analysis can potentially help shed light on key processes in the Mediterranean hydrological cycle. I therefore recommend accepting the paper after addressing my major comments, provided below.

We thank Reviewer 2 for their review and constructive comments. We address them below point by point, together with a description of related changes in our revised manuscript.

[Figure]

Figure R4: Annual mean climatological contributions from advection (left) and divergence (right) to $P - E$ in the Mediterranean region by the stationary terms: (a,b) pure stationary eddies, (c,d) zonally anomalous circulation, and (e,f) zonally anomalous water vapor. Units are millimeters per day.

[Figure]

Figure R5: Annual mean climatological contributions to $P-E$ from horizontal advection of zonally anomalous moisture (that is, the sum of the terms shown panels a and e in FigR4) from ERA5. Units are in millimeters per day

The stated objective of this analysis, to better understand the contributions of time mean and stationary eddies (lines 58–59), is rather incremental. On top of that, the authors do little to delineate their results from previous work. It is therefore not clear what are actual novel findings of the analysis. Further, in several places, it is stated that the results are consistent with previous works, without citing those works. Aside for being bad practice, this further obfuscates the potential novel contributions of the present analysis. In summary, the authors should do a much better job of referencing previous work and clearly stating the novel findings of the present analysis.

We thank the reviewer for this comment, and we agree we could have done a better job at connecting our work to previous studies and at highlighting its novel aspects. The reviewer is right that our work is somewhat incremental: moisture budget analyses both at the global scale and in the Mediterranean region have been performed before based on different reanalysis datasets, including the previous generation of the ECMWF reanalysis (ERA-Interim). Our work, however, goes beyond just showing how previous results are confirmed by the more recent ERA5 (which in itself would be a worthy undertaking), and exposes some novel results. A few among them worth summarizing here are:

- Highlighting persisting artificial trends in hydrological variables in ERA5;

- Clearly exposing the role that stationary eddy moisture fluxes and associated convergence play in the maintenance of net precipitation in the Mediterranean region;

- Quantifying the relative contribution of zonal variations in wind patterns and in moisture (and possible co-variations).

Following these general remarks by Reviewer 2, together with similar comments by Reviewer 1 and specific comments by both reviewers, we have made corresponding changes in the manuscript, both in the Introduction and in the Result Sections. Please see also the reply to the first comment

of Reviewer 1 above. We hope the Reviewers find these changes satisfactory.

It is interesting to note that the periods during which global mean does not equal go along with rapid global warming (Figure 1). does not equal during the rapid global warming from 1979 to the end of the previous century, ending in the 97/8 Niño event. This is followed by a period of weak temperature increase (the so-called 'global warming hiatus') when the residual is small. Then, deviates from again as the rate of global warming increases again after 2012. Can you convince the reader that the global residual of is not due to the moisture storage term in the moisture equation or due to inaccuracies in your methodology? Specifically, one can use Clausius Clapeyron (CC) to demonstrate the former. Under constant relative humidity , we would get where is the CC parameter ( 7%) and is the rate of global warming. Given that the global mean of precipitable water is about 20mm, this yields for a rate of global warming between 1979—2000 of 0.03K/year, justifying the assumption of steady state. It therefore remains to make the case that the residuals are not due to your methodology. For example, due to the use of fewer than available vertical levels, or omitting from the integral near-surface values in regions where surface pressure exceeds 1000hPa. (One way of estimating the integration error may be to compare precipitable water values provided in ERA5 with those derived by integration.) Another potential source of the residual is changes in ice volume. In summary, the authors should convince the reader that the residual of is indeed a feature of the ERA5 reanalysis and not due to their methodology.

We thank the reviewer for this comment, as it allows us to clarify a few important points that should convince the reviewer (and all readers) that the moisture budget residuals and trends both in the global average and over the Mediterranean region are not an artifact of our analysis:

1. In the annual mean, globally averaged $P$ needs to balance globally averaged $E$. This is just a statement of water vapor conservation. There can be trends in both variables, and these trends can be different at the regional scale, but on the global scale, the two terms need to be the same. If they are not, the budget is not closed, that is, there are artificial sinks/sources of moisture in the renalyses, as shown by a number of previous studies (Allan et al., 2020; Hersbach et al., 2020; Mayer et al., 2021). In Fig. R6, we do indeed show that the globally averaged moisture flux convergence (pink line) globally averages to zero (as it should). Additionally, while the storage term, through the dependence of precipitable water on temperature following the CC relationship, can change with seasons, annual mean changes have been shown to be very small in different reanalysis datasets (Mayer et al., 2021; Seager and Henderson, 2013; Trenberth, 1998). This is also evident in Fig. R6, with the green line showing the globally averaged storage term $\frac{\partial}{\partial t} \langle q \rangle$. Note how the magnitude of this term ($\sim 10^{-3}$ mm/day) is about three orders of magnitude smaller than globally averaged $P$ and/or $E$ (Fig. 1 in the manuscript) and how it doesn't feature any significant trend.

2. Together with precipitation and evaporation fluxes, ERA5 publishes vertically integrated moisture fluxes and associated convergence, which is what we use for our analysis of the moisture budget closure, both globally and regionally (Section 2.3 of the manuscript). In other words, there is no vertical integration nor other operations we had to perform to check

[Figure]

Figure R6: Temporal evolution of globally and annually averaged vertically integrated total moisture flux convergence (MFC, pink line) and tendency of vertical integral of water vapor (Tendency, green line) in the $1979 - 2020$ period from ERA5. Units are in millimeters per day.

this closure.

3. Focusing on the moisture budget closure over the Mediterranean region (Figure 2a in the manuscript), we compare the balance between $P - E$ and moisture flux convergence (MFC, again downloaded as a vertically-integrated field from the Copernicus Climate Data Store). The imbalance between $P - E$ and MFC in Fig.2a,b in the manuscript shows that the "regional" budget, is not closed either. Similarly to what is seen on the global average, this residual is not explained by the much smaller tendency of the vertical integral of water vapor (Fig.R7).

We have tried to better emphasize all of these points in our revised manuscript through changes in Section 2 and related subsections.

Given that one potential novelty of this work is to demonstrate the contribution of zonal anomalies in either the humidity or the winds, showing only the immediate Mediterranean region make it difficult to see whether zonally asymmetric terms are related to zonal overturning circulation with links to either the Atlantic or to Asia and Arabia. For example, what is the driver of the negative contribution in the eastern Mediterranean by the dynamic term shown in Fig. 8g? Could this be related to the descending branch of the Indian monsoon (i.e., the so-called Monsoon-Desert mechanism by Rodwell and Hoskins)? Expounding on such processes in the analysis and zonally extending the analysis region may shed light on such potential drivers of the various patterns (e.g., the Indian monsoon, the Persian trough, ventilation of land areas, etc.), which are hardly discussed in the text.

Thanks for this comment. Please see our reply to Reviewer 1's main comment. Following your suggestion, in Fig. R2 and R3 we use a larger domain to show zonal anomalies in wind and moisture patterns. Together with showing these figures in our revised manuscript, we also

[Figure]

Figure R7: Spatial variations of the annual mean climatological tendency of the vertical integral of water vapor in the $1979-2020$ period from ERA5. Units are in millimeters per day.

provide a much more detailed discussion on the influence of these patterns on the net precipitation of our target region, their physical interpretation and connection to previous work (Section 3).

**Comments by line number** 10—13 This sentence is cryptic

Thank you. This sentence has been revised.

48 I would also add Elbaum et al. (2022, "Uncertainty in projected changes in precipitation minus evaporation: Dominant role of dynamic circulation changes and weak role for thermodynamic changes.")

Thank you for this reference. We added it to our discussion.

122 Why is only a subset of the vertical levels used for the vertical integration? Wouldn't this reduce the accuracy of the vertical integration?

The choice of vertical levels was based on a balance between accuracy and computational efficiency. A sub-selection of available vertical levels is common practice in the literature (e.g., D'Agostino and Lionello, 2020; D'Agostino and Lionello, 2017; Minallah and Steiner, 2021; Simpson et al., 2014). To clear any concern, in Fig. R8, we show how vertical integration over 18 levels (1000 975 950 900 850 800 750 700 650 600 550 500 400 300 200 100 50 1 hPa) yields results that are not appreciably different from the ones based on integration over 11 levels.

185—186 This gives the impression that, as in the global mean, there is some constraint under which we would expect to vanish when averaged regionally, which is not the case.

Thanks for this comment. We revised this sentence in the manuscript, to clarify that at the

[Figure]

Figure R8: Annual mean climatological Mediterranean moisture budget in the $1979-2020$ period from ERA5: (a) transient eddies and (b) total stationary eddies using 11 pressure levels for vertical integration (as in the manuscript), and c) transient eddies and d) total stationary eddies using 18 pressure levels for vertical integration. Units are in millimeters per day.

regional scale, a closed moisture budget means that $P - E$ should balance MFC.

213 and elsewhere What previous work? Please specifically cite relevant works

In the revised version, we now provide the relevant references wherever we mention results from previous studies.

245—247 Not sure I agree with this statement. The terms $\nabla \cdot \langle \overline{\mathbf{u}}^* \overline{q}^* \rangle$ and $\nabla \cdot \langle \overline{\mathbf{u}}^* [\overline{q}] \rangle$ generally balance out, and so the term $\nabla \cdot \langle [\overline{\mathbf{u}}] \overline{q}^* \rangle$ would seem to be of the same order as their residual. More generally, the $\nabla \cdot \langle \overline{\mathbf{u}}^* [\overline{q}] \rangle$ term would be related to the zonally asymmetric circulation, whereas the $\nabla \cdot \langle [\overline{\mathbf{u}}] \overline{q}^* \rangle$ term would be related to zonally asymmetric temperature variation. If indeed the latter term is not significant, how does this sit with the alleged immense importance of land-ocean contrasts?

We again refer the reviewer to our response to Reviewer 1's main comment. We agree that we might have underplayed a bit the role of the $\nabla \cdot \langle [\overline{\mathbf{u}}] \overline{q}^* \rangle$ term. We now provide a much more detailed description of the link between zonally anomalous moisture and zonally anomalous circulations, and their co-variations. In particular, we highlight how zonal asymmetries in moisture are less important than zonal asymmetries in vertical motions and affect the region's net precipitation pattern primarily through their horizontal advection by the time mean circulation.

255 Why do the transient eddies dominate the sector mean? Please provide an explanation backed by the relevant references. Jet?

The midlatitude winter storms which propagate along the Atlantic storm track are the primary mechanism sustaining the cold season wet conditions within the region (D'Agostino and Lionello, 2020; Giorgi and Lionello, 2008; Simpson et al., 2015; Zappa et al., 2015). It is the southward shift of the jet to lower latitudes that strengthens the Mediterranean storm track (Flaounas et al., 2022), resulting in net precipitation across the region.

These transient storm systems converge the moisture that originates from the sea over the land regions, resulting in the removal and divergence of moisture from the Mediterranean Sea, and convergence over the land regions. This feature can be clearly seen in Fig.7a in the manuscript, as well: there is a positive net precipitation tendency poleward of 38°N, and lower latitudes within the region, and a negative net precipitation tendency from 32°N to 38°N. Therefore, this term explains much of the $P - E$ pattern within the region in winter, resulting in it dominating the sector-mean hydrological cycle. This is unlike, what happens in summer (JJA): as the jet shifts poleward and strong zonally anomalous descent is established over the region, transient storms weaken and $P - E$ patterns are primarily explained by the different stationary terms. We expanded the discussion in Section 4 to clarify these points.

Figure 4: the mean and stationary terms in Fig. 4e and 4h are nearly identical. Can you explain why? Please comment on this.

Yes! This is indeed one of our key findings: the contribution from the annually averaged mean meridional circulation (Figure 4g in the manuscript) is small, mostly restricted to the southern Mediterranean region. Therefore, the time-mean moisture flux convergence (Fig. 4e) is for the largest part explained by the stationary eddy term (Figure 4h). This is an important result and clarifies how caveats should be taken when talking about the influence of the Hadley cell's descending branch (and it expansion with warming) on the aridity of the Mediterranean region.

317—321 Note that recently, Adam et al.(2023, "Reduced Tropical Climate Land Area Under Global Warming.") showed that over land areas the subtropics are expanding on both their poleward and equatorward edges, and that this expansion is likely driven by thermodynamic drying (reduced evaporative cooling), rather than a dynamic expansion of the tropical overturning circulation.

Many thanks for bringing this interesting article to our attention. We will keep it mind for our future study on changes in the Mediterranean climate. However, it is beyond the scope of our current preprint. Additionally, we do believe the mechanisms invoked in this paper affect tropical land areas, that is latitudes lower than the ones considered here.

**References**

Allan, R. P., Barlow, M., Byrne, M. P., Cherchi, A., Douville, H., Fowler, H. J., Gan, T. Y., Pendergrass, A. G., Rosenfeld, D., Swann, A. L., Wilcox, L. J., and Zolina, O.: Advances in understanding large-scale responses of the water cycle to climate change, Annals of the New York Academy of Sciences, 1472, 49–75, https://doi.org/10.1111/NYAS.14337, 2020.

Cherchi, A., Annamalai, H., Masina, S., and Navarra, A.: South Asian summer monsoon and the eastern Mediterranean climate: The monsoon-desert mechanism in CMIP5 simulations, Journal of Climate, 27, 6877–6903, https://doi.org/10.1175/JCLI-D-13-00530.1, 2014.

Cherchi, A., Annamalai, H., Masina, S., Navarra, A., and Alessandri, A.: Twenty-first century projected summer mean climate in the Mediterranean interpreted through the monsoon-desert mechanism, Climate Dynamics, 47, 2361–2371, https://doi.org/10.1007/S00382-015-2968-4/FIGURES/6, 2016.

D'Agostino, R. and Lionello, P.: The atmospheric moisture budget in the Mediterranean: Mechanisms for seasonal changes in the Last Glacial Maximum and future warming scenario, Quaternary Science Reviews, 241, https://doi.org/10.1016/j.quascirev.2020.106392, 2020.

D'Agostino, R. and Lionello, P.: Evidence of global warming impact on the evolution of the Hadley Circulation in ECMWF centennial reanalyses, Climate Dynamics, 48, 3047–3060, https://doi.org/10.1007/s00382-016-3250-0, 2017.

Flaounas, E., Davolio, S., Raveh-Rubin, S., Pantillon, F., Miglietta, M. M., Gaertner, M. A., Hatzaki, M., Homar, V., Khodayar, S., Korres, G., Kotroni, V., Kushta, J., Reale, M., and Ricard, D.: Mediterranean cyclones: current knowledge and open questions on dynamics, prediction, climatology and impacts, Weather and Climate Dynamics, 3, 173–208, https://doi.org/10.5194/WCD-3-173-2022, 2022.

Giorgi, F. and Lionello, P.: Climate change projections for the Mediterranean region, Global and Planetary Change, 63, 90–104, https://doi.org/10.1016/J.GLOPLACHA.2007.09.005, 2008.

Hersbach, H., Bell, B., Berrisford, P., Hirahara, S., Horányi, A., Muñoz-Sabater, J., Nicolas, J., Peubey, C., Radu, R., Schepers, D., Simmons, A., Soci, C., Abdalla, S., Abellan, X., Balsamo, G., Bechtold, P., Biavati, G., Bidlot, J., Bonavita, M., Chiara, G. D., Dahlgren, P., Dee, D., Diamantakis, M., Dragani, R., Flemming, J., Forbes, R., Fuentes, M., Geer, A., Haimberger, L., Healy, S., Hogan, R. J., Hólm, E., Janisková, M., Keeley, S., Laloyaux, P., Lopez, P., Lupu, C., Radnoti, G., de Rosnay, P., Rozum, I., Vamborg, F., Villaume, S., and Thépaut, J. N.: The ERA5 global reanalysis, Quarterly Journal of the Royal Meteorological Society, 146, 1999–2049, https://doi.org/10.1002/qj.3803, 2020.

Mayer, J., Mayer, M., and Haimberger, L.: Consistency and homogeneity of atmospheric energy, moisture, and mass budgets in ERA5, Journal of Climate, 34, 3955–3974, https://doi.org/10.1175/JCLI-D-20-0676.1, 2021.

Minallah, S. and Steiner, A. L.: Role of the Atmospheric Moisture Budget in Defining the Precipitation Seasonality of the Great Lakes Region, Journal of Climate, 34, 643–657, https://doi.org/10.1175/JCLI-D-19-0952.1, 2021.

Rodwell, M. J. and Hoskins, B. J.: Monsoons and the dynamics of deserts, Quarterly Journal of the Royal Meteorological Society, 122, 1385–1404, https://doi.org/10.1002/QJ.49712253408, 1996.

Seager, R. and Henderson, N.: Diagnostic computation of moisture budgets in the ERA-interim reanalysis with reference to analysis of CMIP-archived atmospheric model data, Journal of Climate, 26, 7876–7901, https://doi.org/10.1175/JCLI-D-13-00018.1, 2013.

Simpson, I. R., Shaw, T. A., and Seager, R.: A diagnosis of the seasonally and longitudinally varying midlatitude circulation response to global warming, Journal of the Atmospheric Sciences, 71, 2489–2515, https://doi.org/10.1175/JAS-D-13-0325.1, 2014.

Simpson, I. R., Seager, R., Shaw, T. A., and Ting, M.: Mediterranean Summer Climate and the Importance of Middle East Topography, Journal of Climate, 28, 1977–1996, https://doi.org/10.1175/JCLI-D-14-00298.1, 2015.

Trenberth, K. E.: Atmospheric Moisture Residence Times and Cycling: Implications for Rainfall Rates and Climate Change, Climatic Change 1998 39:4, 39, 667–694, https://doi.org/10.1023/A:1005319109110, 1998.

Wills, R. C. and Schneider, T.: Stationary eddies and the zonal asymmetry of net precipitation and ocean freshwater forcing, J. Clim., 28, 5115–5133, 2015.

Zappa, G., Hoskins, B. J., and Shepherd, T. G.: The dependence of wintertime Mediterranean precipitation on the atmospheric circulation response to climate change, Environmental Research Letters, 10, 104 012, https://doi.org/10.1088/1748-9326/10/10/104012, 2015.

---

## Author Response (AR2)

**Response to the reviewer's and editor's comments on the manuscript "Revisiting the Moisture Budget of the Mediterranean Region in the ERA5 Reanalysis"**

Roshanak Tootoonchi; Simona Bordoni; Roberta D'Agostino

We thank the anonymous reviewer and the editor for their feedback. In particular, we really appreciate the editor taking the time to review the manuscript and avoiding further delays on the handling of our manuscript. We addressed all reviewers' suggestions in our revised submission, as detailed below in our point-to-point response (in black) to the reviewers' comments (in blue).

**Reviewer 1**

This is my second review of this work, which analyzes the moisture budget of the Mediterranean region using the ERA5 reanalysis. The authors made significant improvements in clarifying the novel contributions of the work, and in providing context for the stationary contributions to the hydrological cycle in the region. I therefore recommend accepting the paper, with some minor comments and suggestions.

Thank you for your positive evaluation and comments!

Sector mean plots (3, 11, A1): Strong land-ocean contrasts are seen in some of the components (transient eddy in particular). Therefore, there should be some sensitivity to the choice of zonal boundaries (10W—40E). For example, the transient eddy sector mean may vary substantially if the sector width would narrow/widen; the authors should address this concern. It might also be helpful to decompose the sector means into land and ocean averages, at least for some of the fields.

Thanks for bringing up this point. We chose our sector as described in the manuscript to be consistent with previous work, such as D'Agostino and Lionello (2020); Giorgi and Lionello (2008); Tuel et al. (2021). We conducted sensitivity studies and found that the emerging zonal-mean patterns are robust to different choices of the box boundaries, which might be more conventionally associated with the Mediterranean region. As an example, in Fig. R1 we show the zonal mean over the 5°W-35°E°N sector (to be compared with Fig. 3 in the revised manuscript). To the extent the chosen range is wide enough that the resulting zonal average is meaningful and representative of most of the Mediterranean region, results remain consistent with what discussed in the manuscript. We also thank the reviewer for suggesting to decompose sector means into land and ocean averages, but we believe that those patterns are sufficiently evident and more meaningful when looking at spatial patterns. We were also reluctant to add more figures (see point below).

Overall, there are 46 panels presented in the main text (not including appendices). At some point, it becomes hard to separate the wheat from the chaff. Here are some suggestions. Figure 4: given that the authors have established that the residuals are small, either Fig. 4c or 4d can be removed. Similarly, Fig. 4g is not very informative, as it is already shown in Fig.3. The same rationale can be applied to Figs. 12 and 13. Generally, using a 2D map to show the sector mean is not a good use of space. Figs. 7b and 8b can be shown as side panels with line plots.

[Figure]

Figure R1: Climatological annual and zonal sector mean moisture budget across the Mediterranean region, with a modified narrower longitude range (5°W-35°E).

We thank the reviewer for their kind suggestion. We decided to condense the information previously conveyed in Figs. 4 and 5 into one figure (Fig. 4 in the updated manuscript). This has been achieved by adopting the same format as in Figs. 12 and 13 for the solstice means of the older manuscript (now Figs. 10 and 11). Please see Fig. R2. Similarly, for Figs. 7 and 8 of the old version, we followed the reviewer's advice and now show zonal averages as line plots on side panels, as shown in Figs. R3 and R4. Along the same lines and following a similar comment by the editor, we also removed Fig. 6 of the older manuscript and now just provide a short qualitative description.

L179 -10E − > 10W
L236 Evaporative body − > source of moisture
Figure 10 caption: day.

Thanks! Done!

**Editor**

The authors have addressed most of the reviewers comments, but based on the re-review of initial reviewer 2 (now reviewer 1) and my own reading and going over the response to initial reviewer 1's concerns , some additional points need to be addressed. Specifically, please address and justify the choice of zonal range for the averaging done in figures 3, 11, A1.

[Figure]

Figure R2: Annual mean climatological moisture budget in the $1979-2020$ period from ERA5: a) $\overline{P} - \overline{E}$, b) monthly mean flow, c) zonal mean flow, d) transient eddies, e) total stationary eddies and its constituent components arising from f) pure stationary eddies, g) transport of zonal mean moisture by the zonally anomalous circulation, and h) transport of zonally anomalous moisture by the zonal mean circulation. Units are millimeters per day.

[Figure]

Figure R3: Annual mean climatological low-level (850 hPa) a) zonally anomalous moisture and b) zonal-mean moisture from ERA5 in the $1979 - 2020$ period. Units are $g\,kg^{-1}$.

[Figure]

Figure R4: Annual mean climatological mid-tropospheric (500 hPa) a) anomalous vertical velocity and b) zonal-mean vertical velocity from ERA5 in the $1979 - 2020$ period. Units are $Pa\,s^{-1}$. Note how here we show negative pressure velocity; hence, positive values (green) denote ascending motion and negative values (brown) denote descending motion.

Many thanks for pointing this out. We now provide references for this choice of latitude-longitude box. Please also see our response to Reviewer 1's first point.

I agree with the reviewer that reducing the number of plots will make the paper easier to follow. Specifically, please consider plotting the sector mean averages alongside the 2-D fields and also maybe marking the averaging region.

Thanks! Please, see our reply to Reviewer 1's second point.

I also found a few points that are confusing that need to be addressed: 1) Figure 6 is not really described in the text, and the comment on line 267 which initially I thought were meant to refer to figures 6 and 7, because figure 6 shows the circulation, but it is also showing the smaller domain.

Thanks you. We indeed removed Fig. 6 as per your suggestion.

2) I am confused by the explanations of the different cancellations of the moisture flux terms between the stationary eddies and the zonal mean moisture advection by the zonally varying flow, as described for example in the response file on page 5 at the top (first full [paragraph starting on line 5). Figure 7 in the manuscript suggests the meridional gradients of the zonally symmetric and zonally varying moisture fields are oppositely signed so the same zonally varying moistions will have opposing meridional advections, which is one cancellation. It is much more confusing to understand this cancellation if we look at the divergence (vertical velocity) where what matters is the sign of the moisture component. Actually, the two terms, divergence times the zonal mean moisture and divergence times the moisture zonal anomaly will only cancel where the zonal moisture anomaly is negative, because the zonal mean moisture has to be positive. The total flux term is teh sum of the advection and the divergence, as you rightly noted, but the physical explanation of the cancelation is done while mixing the advection, the divergence and the total flux terms together.

Yes, you are completely right. Given that the zonally anomalous moisture ($q^*$) varies primarily in the meridional direction, and that its gradients have opposite sign to those of the zonal mean moisture ($[q]$), the same zonally anomalous horizontal motion ($\mathbf{u}^*$) results in opposing meridional advection of $q^*$ (due to the pure stationary term) and meridional advection of $[q]$. In our previous explanation, we focused primarily on the vertical advection (horizontal divergence) term, even if we did acknowledge the important contribution of horizontal advection (which we also show explicitly).

As for the cancellation arising from the vertical advection terms, as you mentioned in your comment, what matters is the sign of the zonally anomalous moisture, given that zonal mean moisture is positive at all latitudes. Hence, the cancellation must arise where we have dry moisture anomaly (negative $q^*$). This is, for instance, what is seen over the Mediterranean Sea itself. We have tried to clarify all of these points in our revised version. Please see lines 310-320.

3) I do not understand the sentence starting on line 310: "In particular..." How can anomalous

descent lead to a moisture flux convergence, unless it is convergence in regions adjacent to the descent. Or do you mean that the divergence in a dry region enhances the initial moisture anomaly ?

As discussed in our reply to your comment above, we have significantly modified the paragraph with a physical explanation of the canceling tendencies between the pure stationary term and the term arising from transport of zonally averaged moisture by the zonally anomalous circulation. The confusing sentence now reads: "The drying effect of the zonally anomalous divergent circulation is reduced by the pure stationary term in regions of reduced moisture availability ($\overline{q}_{850}^* < 0$)". We hope this new sentence, and related discussion, clarifies any pending confusion.

4) Response to the comment about the local Hadley cell paper by Li et al- I think what the reviewer meant is that the Hadley cell descent mechanism often invoked for the Med. being a dry region (along with the Hadley cell expansion being a main cause for future drying) is according to your results not the leading explanation, because the stationary waves are much more dominant than teh zonal mean overturning term. However, if you take a more lax definition of the Hadley circulation as the overturning divergent circulation, then can you say that the Hadley circulation descent is a leading drying process for the Mediterranean? It is possible to discuss this qualitatively, by simply looking at the Li et al papers or the papers they reference to see if the Mediterranean is indeed a mean ascent region in the annual mean, and then in the solstice seasons. Making this quantitative requires I think dividing the fluxes into rotational and divergent, and maybe regressing on the regional Hadley cell, and I agree is beyond the scope of your paper.

Motivated by this comment, we have conducted preliminary analyses on the regional overturning circulation. That is, we have divided the flow into rotational and divergent components and computed the longitudinally varying meridional streamfunction from the divergent component, following, for instance, Zhang and Wang (2013). The vertical velocity associated with this regional meridional overturning can then be computed as the streamfunction meridional derivative. The result is shown in Fig. R5. Please note that, because of ongoing issues with the Climate Data Store, we downloaded only a subset of the years used in this study (2000-2020 rather than 1979-2020). We, however, believe this result will not change significantly if we extend the analysis to all years. We see that while there is indeed descent over the Mediterranean region, it is weak and much smaller than the total descent (Fig.R4). The same holds true for the solstice seasons. This leads us to conclude that the regional meridional overturning does not capture all zonal asymmetries of the mean flow and their influence on the net precipitation of the Mediterranean region. We have added the discussion of this preliminary analysis on Lines 439-449 of the revised manuscript.

[Figure]

Figure R5: Annual mean climatological mid-tropospheric (500 hPa) vertical velocity $Pa\,s^{-1}$ associated with the regional overturning streamfunction. Note how here we show negative pressure velocity; hence, positive values (green) denote ascending motion and negative values (brown) denote descending motion.

**References**

D'Agostino, R. and Lionello, P.: The atmospheric moisture budget in the Mediterranean: Mechanisms for seasonal changes in the Last Glacial Maximum and future warming scenario, Quat. Sci. Rev., 241, 106 392, 2020.

Giorgi, F. and Lionello, P.: Climate change projections for the Mediterranean region, Glob. Planet. Change, 63, 90–104, 2008.

Tuel, A., O'Gorman, P. A., and Eltahir, E. A.: Elements of the Dynamical Response to Climate Change over the Mediterranean, Journal of Climate, 34, 1135–1146, https://doi.org/10.1175/JCLI-D-20-0429.1, 2021.

Zhang, G. and Wang, Z.: Interannual Variability of the Atlantic Hadley Circulation in Boreal Summer and Its Impacts on Tropical Cyclone Activity, Journal of Climate, 26, 8529 – 8544, https://doi.org/10.1175/JCLI-D-12-00802.1, 2013.

---

## Author Response (AR3)

**Response to the editor's comments on the manuscript "Revisiting the Moisture Budget of the Mediterranean Region in the ERA5 Reanalysis"**

Roshanak Tootoonchi; Simona Bordoni; Roberta D'Agostino

We thank the editor for taking the time to review the manuscript again and their feedback. We addressed all suggestions in our revised submission, as detailed below in our point-to-point response (in black) to the editor's comments (in blue).

**Editor**

The authors have addressed the comments by the reviewer 1 satisfactorly, however I disagree with the conclusion re the local Hadley cell interpretation. It is great that you have calculated the irrotational overturning circulation, but my interpretation of the comparison between figures R5 to R4 is that about 66 % of the downwelling center over the south eastern Mediterranean is part of the irrotational flow (the peak values are about 0.04 Pa/sec in R5 vs about 0.06 Pa/sec in figure R5). Please correct me if I am interpreting things wrong, but if not, then I would revise the text on lines 440-444 to something like: " Preliminary analyses we have conducted using this framework, suggest that both the localized overturning circulation and the stationary wave associated descending motions play a role, with the contribution of the overturning circulation to the peak descent region (over Egypt) being about two thirds. In other words, even if a more generalized localized view of the Hadley circulation is taken into account, it does not capture all zonal asymmetries of the mean flow and their influence on the net precipitation of the Mediterranean, and stationary waves need to be taken into account."

Thank you for your comment. It is indeed true that over the southeastern Mediterranean (Egypt, Libya), the contribution from the regional overturning streamfunction is considerable. Over the rest of the Mediterranean, however, say Italy or Turkey, that is not the case. We took your suggestion into account and modified the paragraph accordingly.

Additional private note (visible to authors and reviewers only): Some minor comments which I also noted: Should the work of Reiter et al / Galanti et al (Yohai Kaspi's group) also be referenced in this context of a regional Hadley circulation?

Thanks for the suggestion. The references are added!

line 323 - add reference to figure 7 which is discussed right afterwards "As will now be shown in fig 7, the spatial patterns of this term cannot simply be explained in terms of the divergence of zonally anomalous moisture by the zonally averaged vertical motion" Or: "AImportantly, the spatial patterns of this term cannot simply be explained in terms of the divergence of zonally anomalous moisture by the zonally averaged vertical motion (fig 7ef)"

Thanks! Done!

remove "that" from line 336 (I think)

Thanks for the suggestion. However, we believe the sentence is correct in its current form.